# ITERATIVE $\alpha$-(DE)BLENDING: LEARNING A DETERMINISTIC MAPPING BETWEEN ARBITRARY DENSITIES

## ABSTRACT

We present a learning method that produces a mapping between arbitrary densities, such that random samples of a density can be mapped to random samples of another. In practice, our method is similar to deterministic diffusion processes where samples of the target density are blended with Gaussian noise. The originality of our approach is that, in contrast to several recent works, we do not rely on Langevin dynamics or score-matching concepts. We propose a simpler take on the topic, which is based solely on basic sampling concepts. By studying blended samples and their posteriors, we show that iteratively blending and deblending samples produces random paths between arbitrary densities. We prove that, for finite-variance densities, these paths converge towards a deterministic mapping that can be learnt with a neural network trained to deblend samples. Our method can thus be seen as a generalization of deterministic denoising diffusion where, instead of learning to denoise Gaussian noise, we learn to deblend arbitrary data.

**We provide a short video overview of the paper in our supplementary material.**

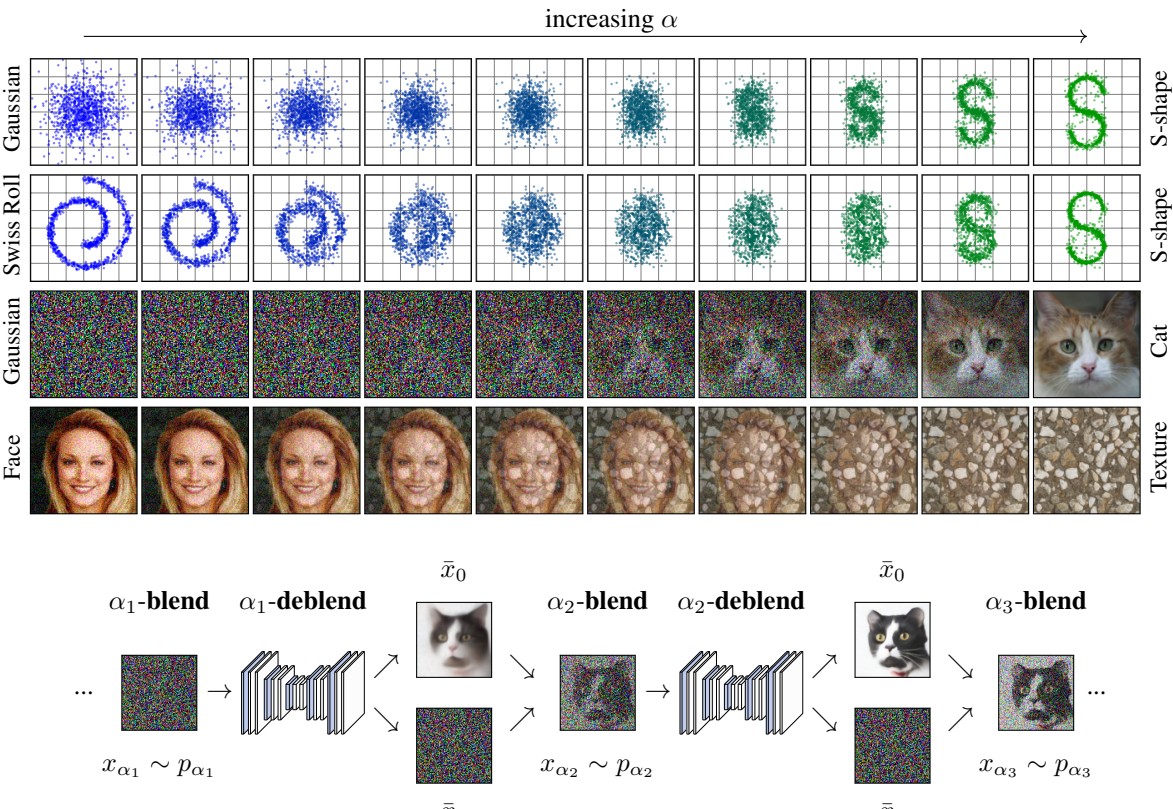

Figure 1: **Iterative $\alpha$-blending and deblending.** We train a neural network to deblend blended inputs. By deblending and reblending iteratively we obtain a mapping between arbitrary densities.

## 1 INTRODUCTION

Diffusion models have recently become one of the most popular generative modeling tools (Ramesh et al., 2022). They have outperformed state-of-the-art GANs (Karras et al., 2020; 2021) and been applied to many applications such as image generation (Rombach et al., 2021; Dhariwal & Nichol, 2021), image processing (Saharia et al., 2021; Kawar et al., 2022; Whang et al., 2022), text-to-image (Saharia et al., 2022b), video (Ho et al., 2022) or audio (Kong et al., 2020).

**First, there were stochastic diffusion models...** These diffusion models have in common that they can be formulated as a *Stochastic Differential Equations* (SDEs) (Song et al., 2021b) such as *Langevin dynamics*. Langevin's equation models a random walk that obeys a balance between two operations related to Gaussian noise: increasing noise by adding more noise and decreasing noise by climbing the gradient of the log density. Increasing noise performs large steps but puts the samples away from the true density. Decreasing noise projects the samples back on the true density. Carefully tracking and controlling this balance allows to perform efficient random walk and provides a sampling procedure for the true density. This is the core of denoising diffusion approaches. Noise Conditional Score Networks (NCSNs) (Song & Ermon, 2019; 2020) use Langevin's equation directly by leveraging the fact that the score (the gradient of the log density in Langevin's equation) can be learnt via a denoiser when the samples are corrupted with Gaussian noise (Vincent, 2011). Denoising Diffusion Probabilistic Models (DDPMs) (Ho et al., 2020; Nichol & Dhariwal, 2021) use a Markov chain formalism with a Gaussian prior that provides an SDE similar to Langevin dynamics where the score is also implicitly learnt with a denoiser.

**...then came deterministic diffusion models.** Langevin's SDEs variants describe an equilibrium between noise injection and noise removal. Nullifying the noise injection in these SDEs yields *Ordinary Differential Equations* (ODEs), also called *Probability Flow ODEs* Song et al. (2021b), that simply describe the deterministic trajectory of a noisy sample projected back in the true density. For instance, Denoising Diffusion Implicit Models (DDIMs) (Song et al., 2021a) are the ODE variants of DDPMs. These ODEs provide a smooth deterministic mapping between the Gaussian noise density and the true density. Deterministic diffusion models been motivated recently because an ODE requires much fewer solver iterations than its SDE counterpart. Furthermore, a deterministic mapping presents multiple practical advantages because samples are uniquely determined by their prior Gaussian noise, can be interpolated via the Gaussian noise, etc.

**Is there a simpler approach to deterministic diffusion?** The point of the above story is that, in the recent line of work on diffusion models, stochastic diffusion models came *first* and deterministic diffusion models came *after*, framed as special cases of the stochastic ones. They hence inherited the underlying mindset and mathematical framework. As a result, advanced concepts such as Langevin dynamics, score matching, how they relate to Gaussian noise, etc. appear to be necessary background to grasp recent deterministic diffusion models. We argue that this is a significant detour to something that can be framed in a much simpler and more general way. We propose a fresh take on deterministic diffusion with another mindset, using only basic sampling concepts.

• We derive a deterministic diffusion-like model based on the sampling interpretation of blending and deblending. We call it Iterative $\alpha$-(de)Blending (IADB) in reference to the Computer Graphics $\alpha$-blending technique that composes images with a transparency parameter Porter & Duff (1984). Our model defines a mapping between arbitrary densities (of finite-variance).

• We show that when the initial density is Gaussian, the mappings defined by IADB are exactly the same as the ones defined by DDIM (Song et al., 2021a). On the theoretical side, our model can thus be seen as a generalization of DDIM to arbitrary sampling densities rather than just Gaussian. Furthermore, our alternative derivation leads to a more numerically stable sampling formulation. Our experiments show that IADB consistently outperforms DDIM in terms of final FID on several datasets and is more stable with small number of steps in the sampling stage.

• We explore the generalization to arbitrary non-Gaussian densities provided by our model. We report that, although this generalization seems promising on the theoretical side, the application possibilities were disappointing in our experiments in image generation. We found that sampling with non-Gaussian densities can significantly lower the quality of the generated samples and that the mappings are not always interesting for image processing applications.

## 2 A DETERMINISTIC MAPPING BETWEEN ARBITRARY DENSITIES

We consider two densities $p_0, p_1 : \mathbb{R}^d \to \mathbb{R}^+$ represented respectively by the red triangle and the green square in Figure 2. Our objective is to define a deterministic mapping such that i.i.d. samples $x_0 \sim p_0$ passed through the mapping produce i.i.d. samples $x_1 \sim p_1$.

### 2.1 BLENDING AND DEBLENDING AS SAMPLING OPERATIONS

**(a) $\alpha$-blending.** We call $p_\alpha$ the density of the blended samples $x_\alpha = (1 - \alpha) x_0 + \alpha x_1$ obtained by blending random samples $(x_0, x_1) \sim p_0 \times p_1$ with a blending parameter $\alpha \in [0, 1]$. This is illustrated in Figure 2-(a).

**(b) $\alpha$-deblending.** We call $\alpha$-deblending the inverse sampling operation, *i.e.* generating random $x_0$ and $x_1$ from the initial densities that could have been $\alpha$-blended into a point $x$, as shown in Figure 2-(b). More formally, it means generating random *posterior samples* $(x_0, x_1)_{|(x,\alpha)} \sim (p_0 \times p_1)_{|(x,\alpha)}$. Note that we never use these posteriors samples in practice, we use them only for the derivation of our method.

**Proposition 1.** *If $x \in \mathbb{R}^d$ is a **fixed** point, the posteriors samples $(x_0, x_1)_{|(x,\alpha)} \sim (p_0 \times p_1)_{|(x,\alpha)}$ are distributed in the posterior densities. However, if $x_\alpha \sim p_\alpha$ is a **random** sample, the posteriors samples are distributed in the initial densities: $(x_0, x_1)_{|(x_\alpha \sim p_\alpha, \alpha)} \sim (p_0 \times p_1)$.*

**Proof.** It follows directly from the law of total probability. We provide more details in Appendix A.

**(c) $\alpha$-(de)blending.** Let's consider two blending parameters $\alpha_1, \alpha_2 \in [0, 1]$. Using the previous proposition, we can chain $\alpha_1$-deblending and $\alpha_2$-blending to map a random sample $x_{\alpha_1} \sim p_{\alpha_1}$ to a random sample $x_{\alpha_2} \sim p_{\alpha_2}$. Indeed, by sampling posteriors for a random sample $x_{\alpha_1} \sim p_{\alpha_1}$, we obtain random samples $(x_0, x_1) \sim (p_0 \times p_1)$ from the initial densities, and blending them with parameter $\alpha_2$ provides a random sample $x_{\alpha_2} \sim p_{\alpha_2}$. This is illustrated in Figure 2-(c).

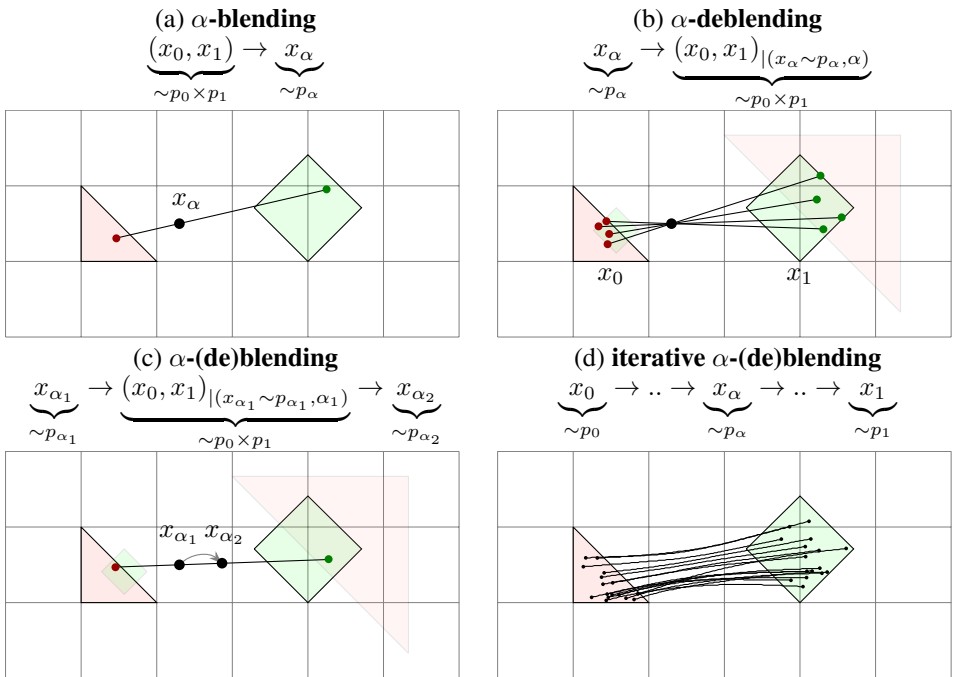

Figure 2: **Blending and deblending as sampling operations.**

## 2.2 ITERATIVE $\alpha$-(DE)BLENDING (IADB)

We introduce Iterative $\alpha$-(de)Blending (IADB), an iterative algorithm that can be implemented stochastically or deterministically. Our main result is that both variants converge towards the same limit, which yields a deterministic mapping between the densities $p_0$ and $p_1$ shown in Figure 2-(d).

**Algorithm 1: iterative $\alpha$-(de)blending (stochastic).** Let's consider a number of iterations $T$ and evenly distributed blending parameters $\alpha_t = t/T, t = \{0, .., T\}$). This algorithm creates a sequence $(x_{\alpha_t} \sim p_{\alpha_t}, t = \{0, .., T\})$ that starts with a random sample $x_0 \sim p_0$ and ends with a random sample $x_{\alpha_T} = x_1 \sim p_1$ by applying $\alpha$-(de)blending iteratively. In each iteration, $x_{\alpha_t} \sim p_{\alpha_t}$ is $\alpha_t$-deblended by sampling random posteriors, which are sampled and $\alpha_{t+1}$-blended again to obtain a new sample $x_{\alpha_{t+1}} \sim p_{\alpha_{t+1}}$. End-to-end, this algorithm provides a stochastic mapping between samples $x_0 \sim p_0$ and samples $x_1 \sim p_1$.

**Algorithm 2: iterative $\alpha$-(de)blending (deterministic).** This algorithm is the same as Algorithm 1 except that, in each iteration, the random posteriors samples are replaced by their expectations. The algorithm is thus not stochastic but deterministic.

**Theorem 1: convergence of iterative $\alpha$-(de)blending.** *If $p_0$ and $p_1$ are Riemann-integrable densities of finite variance, the sequences computed by Algorithm 1 and Algorithm 2 converge towards the same limit as the number of steps $T$ increases, i.e. for any $\alpha \in [0, 1]$ we have*

$$\lim_{T \to \infty} x_\alpha \text{ computed by Algorithm 1}(x_0, T) = \lim_{T \to \infty} x_\alpha \text{ computed by Algorithm 2}(x_0, T). \quad (1)$$

**Proof.** We detail this proof in Appendix B. Intuitively, in each iteration, Algorithm 1 makes a small step $\Delta x_\alpha = (x_1 - x_0) \Delta \alpha$ along the segment given by random posterior samples. As the number of iterations increases, many small random steps average out, and the infinitesimal steps are described by an ODE that involves the expected posteriors like in Algorithm 2:

$$\mathrm{d}x_\alpha = \left( \bar{x}_{1|(x_\alpha, \alpha)} - \bar{x}_{0|(x_\alpha, \alpha)} \right) \mathrm{d}\alpha. \quad (2)$$

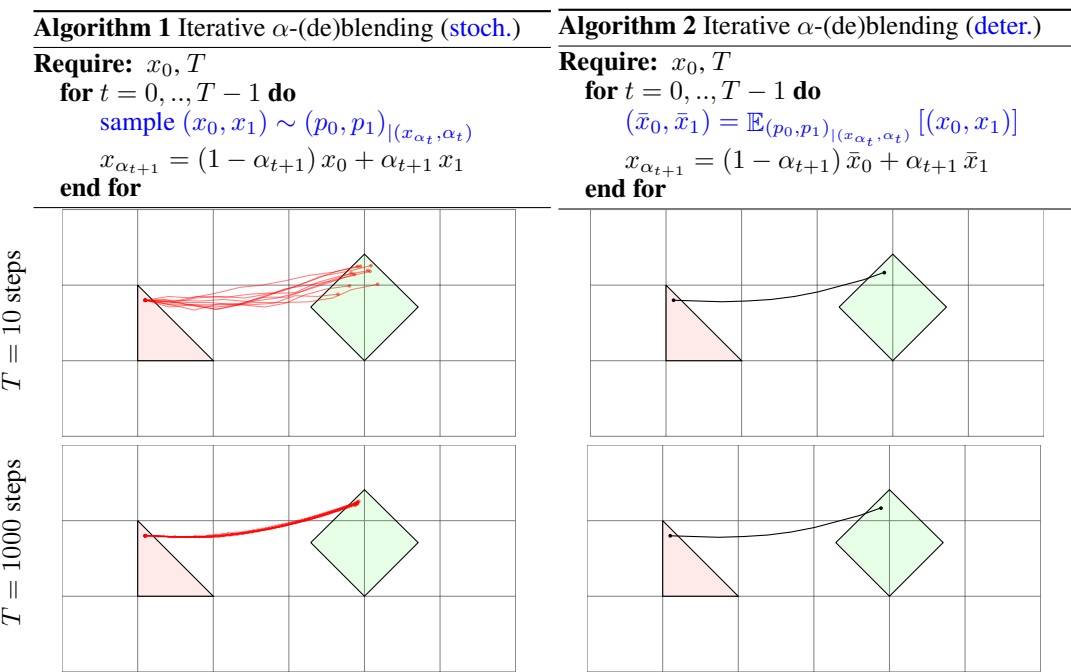

Figure 3: Both algorithms step iteratively by moving the samples along segments defined by their posterior densities. The difference is that Algorithm 1 uses segments between random posterior samples, which creates stochastic paths, while Algorithm 2 uses the segment between the average of the posterior samples, which creates deterministic paths. As the number of steps $T$ increases, the randomness of the stochastic paths averages out and they converge towards the deterministic paths.

## 3 LEARNING ITERATIVE $\alpha$-(DE)BLENDING

In this section, we explain how to use iterative $\alpha$-(de)blending in a machine learning context, where we train a neural network $D_\theta$ to predict the average posterior samples used in Algorithm 2.

### 3.1 VARIANT FORMULATIONS OF ITERATIVE $\alpha$-(DE)BLENDING

A direct transposition of Algorithm 2 means learning the averages of both posterior samples $\bar{x}_0$ and $\bar{x}_1$. However, one is implicitly given by the other one such that it is not necessary to learn both and variants of Alg. 2 are possible. The fact that multiple, theoretically equivalent, variants are possible is pointed out by Salimans & Ho (2022). However, they are not equivalent in practice. In Table 1, we summarize four variants derived in Appendix C and compare their practical properties. Variant (a) is the vanilla transposition of Algorithm 2. It is highly unstable because instead of moving the current sample $x_{\alpha_t}$, the new sample $x_{\alpha_{t+1}}$ is plainly recomputed from the outputs of the neural network, such that its residual learning errors accumulate in each step. The larger the number of steps $T$, the more this variant diverges. Variants (b) and (c) consist of learning either only $\bar{x}_0$ or $\bar{x}_1$. The sampling suffers from numerical stability near respectively $\alpha_t = 0$ and $\alpha_t = 1$ because of the respective divisions by $\alpha_t$ and $1 - \alpha_t$. We recommend using variant (d) that consists of learning the average difference vector $\bar{x}_1 - \bar{x}_0$. It is a direct transposition of the ODE defined in Equation 2. This variant updates the current samples in each iteration without any division. We found it to be the most stable variant for both training and sampling.

| (a) **learn $\bar{x}_0$ and $\bar{x}_1$** | (b) **learn only $\bar{x}_0$** | (c) **learn only $\bar{x}_1$** | (d) **learn $\bar{x}_1 - \bar{x}_0$** |
|---|---|---|---|
| $(\bar{x}_0, \bar{x}_1) = D_\theta\left(x_{\alpha_t}, \alpha_t\right)$ | $\bar{x}_0 = D_\theta\left(x_{\alpha_t}, \alpha_t\right)$ | $\bar{x}_1 = D_\theta\left(x_{\alpha_t}, \alpha_t\right)$ | $\bar{x}_1 - \bar{x}_0 = D_\theta\left(x_{\alpha_t}, \alpha_t\right)$ |
| $x_{\alpha_{t+1}} =$ | $x_{\alpha_{t+1}} = \bar{x}_0 +$ | $x_{\alpha_{t+1}} = \bar{x}_1 +$ | $x_{\alpha_{t+1}} = x_{\alpha_t} +$ |
| $(1 - \alpha_{t+1})\,\bar{x}_0 + \alpha_{t+1}\bar{x}_1$ | $\frac{\alpha_{t+1}}{\alpha_t}\left(x_{\alpha_t} - \bar{x}_0\right)$ | $\frac{(1-\alpha_{t+1})}{(1-\alpha_t)}\left(x_{\alpha_t} - \bar{x}_1\right)$ | $(\alpha_{t+1} - \alpha_t)\left(\bar{x}_1 - \bar{x}_0\right)$ |
| unstable | unstable when $\alpha_t \to 0$ | unstable when $\alpha_t \to 1$ | stable |

Table 1: **Variant formulations of iterative $\alpha$-(de)blending** (equivalent in theory, not in practice).

### 3.2 TRAINING AND SAMPLING

Following variant (d) of Table 1, we train the neural network $D_\theta$ to predict the average difference vector between the posterior samples. Our learning objective is defined by

$$\min_\theta \quad \mathbb{E}_{\alpha, x_\alpha}\left[\left\|D_\theta\left(x_\alpha, \alpha\right) - \mathbb{E}_{x_{0|(x_\alpha, \alpha)}, x_{1|(x_\alpha, \alpha)}}\left[x_{1|(x_\alpha, \alpha)} - x_{0|(x_\alpha, \alpha)}\right]\right\|^2\right]. \tag{3}$$

Note that minimizing the $l^2$ to the average of a distribution is equivalent to minimizing the $l^2$ to all the samples of the distribution. We obtain the equivalent objective

$$\min_\theta \quad \mathbb{E}_{\alpha, x_\alpha, x_{0|(x_\alpha, \alpha)}, x_{1|(x_\alpha, \alpha)}}\left[\left\|D_\theta\left(x_\alpha, \alpha\right) - \left(x_{1|(x_\alpha, \alpha)} - x_{0|(x_\alpha, \alpha)}\right)\right\|^2\right]. \tag{4}$$

Finally, as explained in Section 2.1, sampling $x_\alpha \sim p_\alpha$ and $(x_{0|(x_\alpha, \alpha)}, x_{1|(x_\alpha, \alpha)})$ in this order is equivalent to sampling $x_0 \sim p_0$ and $x_1 \sim p_1$ and blending them to obtain $x_\alpha \sim p_\alpha$. We obtain our final learning objective

$$\min_\theta \quad \mathbb{E}_{\alpha, x_0, x_1}\left[\left\|D_\theta\left((1 - \alpha)\,x_0 + \alpha\,x_1, \alpha\right) - (x_1 - x_0)\right\|^2\right], \tag{5}$$

which we use to optimize $\theta$ in Algorithm 3. Finally, in Algorithm 4, we iteratively map samples $x_0 \sim p_0$ to samples $x_1 \sim p_1$ in the same way as in Algorithm 2 where we use the neural network $D_\theta$ to obtain the average posterior difference.

| **Algorithm 3** Training | **Algorithm 4** Sampling |
|---|---|
| **Require:** $x_0 \sim p_0, x_1 \sim p_1, \alpha \in [0, 1]$ | **Require:** $x_0, T$ |
| $\quad x_\alpha = (1 - \alpha)\,x_0 + \alpha\,x_1$ | $\quad$ **for** $t = 0, .., T - 1$ **do** |
| $\quad l = \|D_\theta\left(x_\alpha, \alpha\right) - (x_1 - x_0)\|^2$ | $\quad\quad x_{\alpha_{t+1}} = x_{\alpha_t} + (\alpha_{t+1} - \alpha_t)\,D_\theta\left(x_{\alpha_t}, \alpha_t\right)$ |
| $\quad$ backprop from $l$ and update $\theta$ | $\quad$ **end for** |

# 4 EXPERIMENTS WITH ANALYTIC DENSITIES

**Mapping 1D densities.** In Figure 4, we experiment with analytic 1D densities where the expectation $\bar{x}_1 - \bar{x}_0$ can be computed analytically rather than being learnt by a neural network $D_\theta$. The experiment confirms that the analytic version matches the reference and that the neural network trained with the $l_2$ approximates the same mapping. We also tested training the neural network with the $l_1$, which makes the neural network approximate the median of $x_1 - x_0$ rather than its average. The resulting mapping does not match the reference. This confirms that learning the average via $l_2$ training is a key component of our model, as explained in Section 3.2.

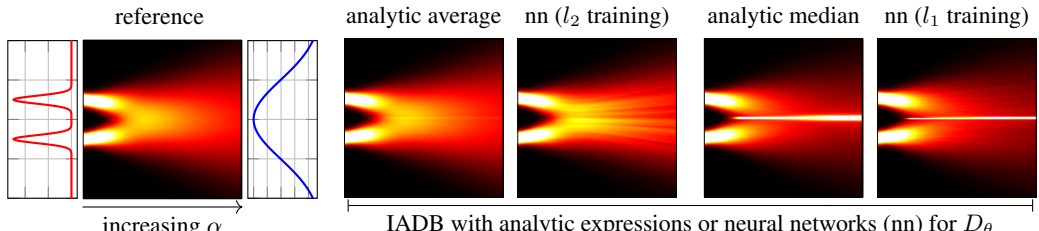

Figure 4: We map a bi-Normal distribution with modes $\mu_1 = -0.5$ and $\mu_2 = 0.5$ with $\sigma_{1|2} = 0.1$ (in red) to a Normal distribution of unit variance (in blue). The reference shows the intermediate blended densities $p_\alpha$ obtained by analytically convolving both densities. The other densities are the histograms of the samples $x_\alpha$ computed in Algorithm 4 using either analytic expressions or neural networks (nn) for $D_\theta$. The neural network is a MLP with 5 hidden layers of 64 filters.

**Mapping 2D densities.** Figure 5 shows that the intermediate blended densities $p_\alpha$ computed by our mapping match the reference blended densities. Figure 6 shows how our algorithm maps the samples of $p_0$ to samples of $p_1$.

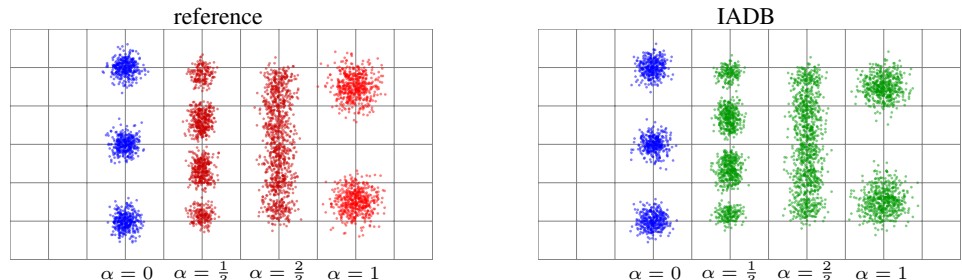

Figure 5: We display blended samples $x_\alpha \sim p_\alpha$ (red) and samples $x_\alpha$ computed by Algorithm 4 using a MLP with 5 hidden layers of 64 filters for $D_\theta$ (green).

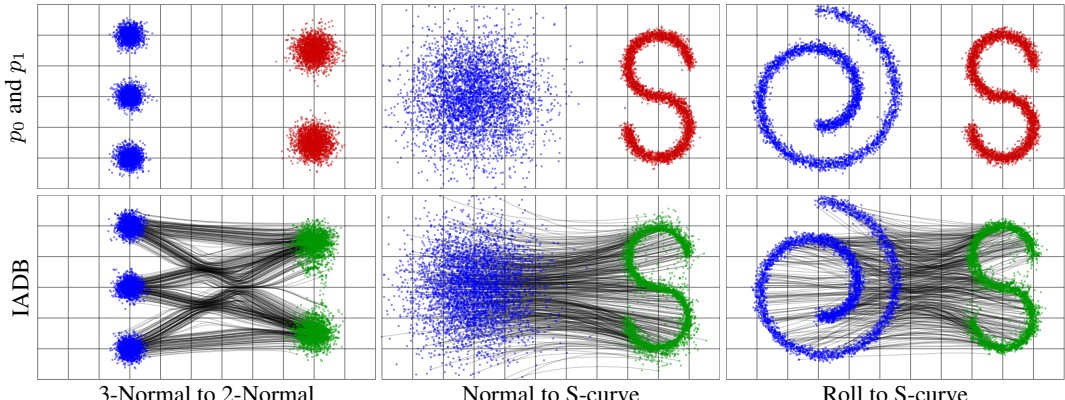

Figure 6: We show samples of the reference densities $p_0$ and $p_1$ and the mapping computed by Algorithm 4 using a MLP with 5 hidden layers of 64 filters for $D_\theta$. The final samples computed by the algorithm (green) match the reference samples $x_1 \sim p_1$ (red).

## 5 RELATION TO DDIM

In this section, we explain that IADB can be framed as a more stable generalization of DDIM (Song et al., 2021a) that enjoys the possibility of using non-Gaussian densities for $p_0$.

**Proposition 2.** *If $p_0$ is a Gaussian density, IADB and DDIM define the same deterministic mapping.*

**Proof.** In Appendix D we show that with a simple change of parameterization, the update rule of DDIM is exactly the variant (b) in our Table 1.

**Experimental comparison of IADB and DDIM.** In Figure 7, we experiment under the same conditions (architecture, training time, 1st-order solver, uniform schedule, Gaussian $p_0$) and measure the FID score (Heusel et al., 2017) for varying number of sampling steps on 3 image datasets: LSUN Bedrooms (64x64), CelebA (64 x 64) and AFHQ Cats(128x128) for 120 hours of training. We use a U-Net architecture(from Diffusers library[1]). We observe a consistently better performance of IADB compared to DDIM. This might be due to our numerically stable formulation explained in Section 3.1. The formulation generally used in DDIM corresponds to the variant (b) presented in Table 1: they train a denoiser to predict the Gaussian noise present in the noisy image samples, *i.e.* their model learns to predict $\bar{x}_0$. However, we explain that this variant makes the sampling less stable because of the division near 0. As a matter of fact, in their implementation, the sampler starts at some $\epsilon > 0$ precisely to avoid dividing by 0. Our variant (d) does not suffer from this problem. Another possibility is that the learning objective defined by variant (d) provides a better optimization landscape than variant (b). For instance, the effort for learning $\bar{x}_0$ is likely imbalanced in $\alpha$ while the effort for learning $\bar{x}_1 - \bar{x}_0$ is likely more balanced over $\alpha$.

**Discussion.** The Langevin/score-matching approach puts the emphasis on the fact that the gradient of the log density in Langevin's equation can be learnt via $\bar{x}_0$ when $p_0$ is Gaussian Vincent (2011). This mindset naturally leads to variant (b) in Table 1. In contrast, the derivation of IADB emphasizes that $x_\alpha$, $\bar{x}_0$ and $\bar{x}_1$ are aligned and thus that all the four variants of Table 1 are possible.

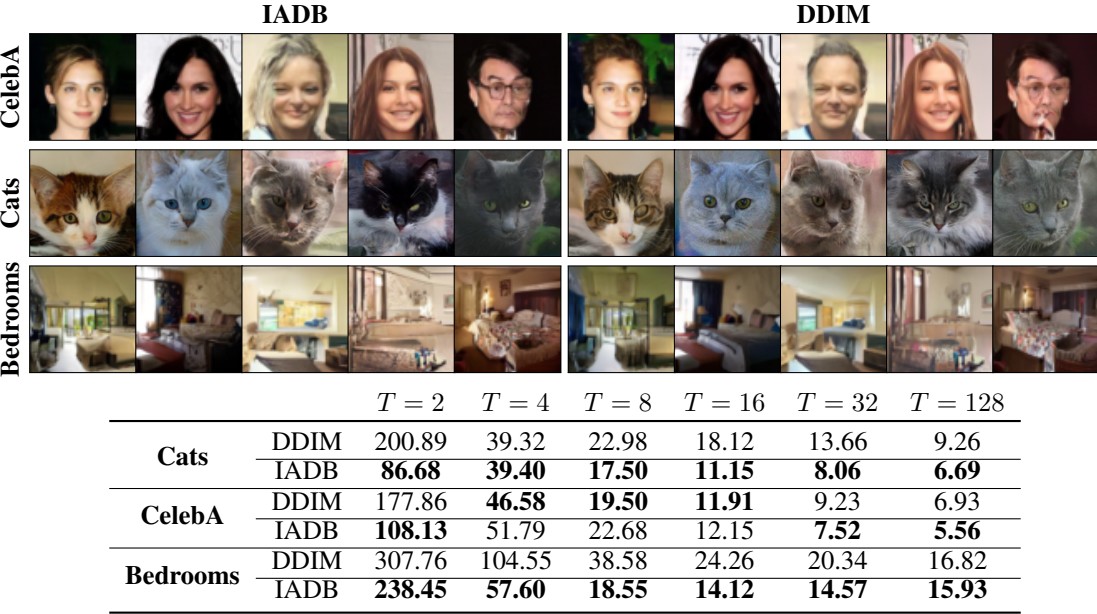

|  |  | $T=2$ | $T=4$ | $T=8$ | $T=16$ | $T=32$ | $T=128$ |
|---|---|---|---|---|---|---|---|
| **Cats** | DDIM | 200.89 | 39.32 | 22.98 | 18.12 | 13.66 | 9.26 |
|  | IADB | **86.68** | **39.40** | **17.50** | **11.15** | **8.06** | **6.69** |
| **CelebA** | DDIM | 177.86 | **46.58** | **19.50** | **11.91** | 9.23 | 6.93 |
|  | IADB | **108.13** | 51.79 | 22.68 | 12.15 | **7.52** | **5.56** |
| **Bedrooms** | DDIM | 307.76 | 104.55 | 38.58 | 24.26 | 20.34 | 16.82 |
|  | IADB | **238.45** | **57.60** | **18.55** | **14.12** | **14.57** | **15.93** |

Figure 7: **Comparing IADB and DDIM.** We use the same Gaussian noise to sample images with IADB and DDIM. We obtain very close images because the underlying theoretical mappings are the same. IADB achieves better FID scores w.r.t. the number of steps $T$ than DDIM most of the time.

---

[1]https://github.com/huggingface/diffusers

# 6 (DISAPPOINTING) EXPERIMENTS WITH ARBITRARY IMAGE DENSITIES

**(Disappointing) sampling quality.** In the experiment of Figure 8, we use IADB to compute mappings between different image datasets, *i.e.* in contrast to the experiment of Figure 7, we use real images rather than Gaussian noise to sample other images. Note that, in contrast to the Gaussian density, the implicit density represented by an image dataset might not be Riemann integrable, for instance if the images are distributed on a lower-dimensional manifold. To make sure that our theorem applies, we regularize $p_0$ by applying a little amount of noise to the images. In theory, with this regularization, IADB is proven to produce a correct sampling of $p_1$ regardless of $p_0$. However, in practice, we observe that the qualitative performance of the mapping is significantly lower than with Gaussian noise. This might be because we use an architecture that has been designed specifically for denoising Gaussian noise or because denoising noise might be fundamentally simpler than deblending arbitrary images. In any case, our takeaway is that an experimental set up that works well with Gaussian noise does not necessarily transpose successfully to other densities.

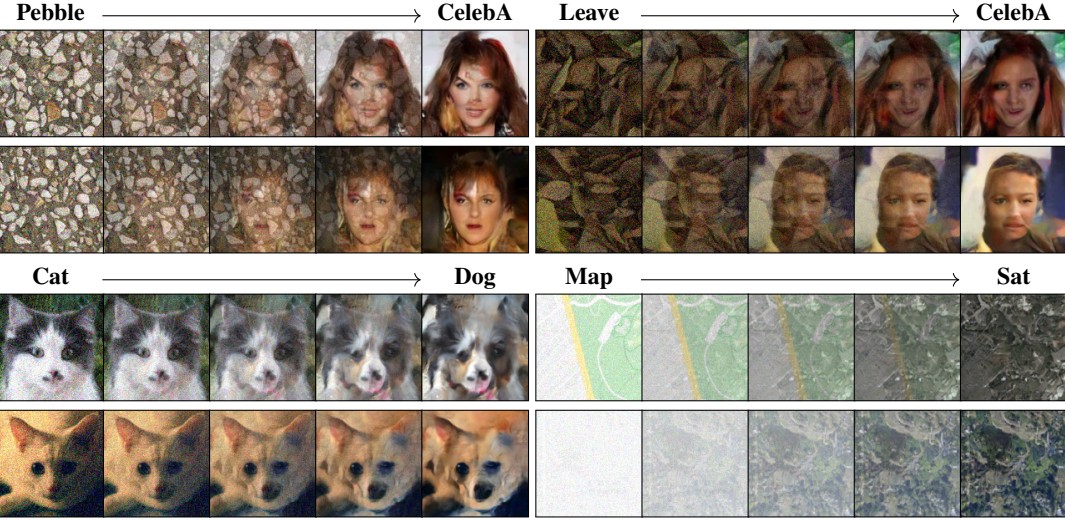

Figure 8: **Sampling with non-Gaussian densities.** We use the same experimental set up as in Figure 7 except that we replace the Gaussian noise by an image database.

**(Disappointing) mappings.** For some applications, we wish that IADB would learn a "meaningful" mapping between $p_0$ and $p_1$. Unfortunately, this is not always the case. Indeed, the theory predicts that the mapping will produce a valid sampling of $p_1$ using $p_0$ but not that the mapping will be what a human user expects. For instance, Figure 9 shows a result where IADB was trained to learn a mapping between corrupted images and clean images and we wish that the mapping would behave like an image restoration process. Unfortunately, although IADB effectively learns to sample clean images using corrupted ones, the mapping does not produce a valid restoration because the clean images do not always resemble the corrupted ones.

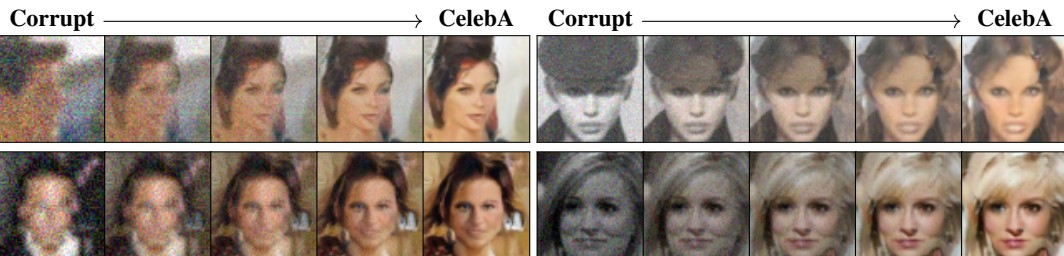

Figure 9: **Image restoration with IADB.** In this experiment, we use IADB to map corrupted images to clean images. (left) The corruption is a downscaling+noise. (right) The corruption is a decolorization+noise. Ideally, the mapping learnt by IADB would restore the corrupted image. Unfortunately, the mapping creates a clean image but it does not match the corrupted one anymore.

## 7 RELATED WORK

**Stochastic Differential Equations (SDEs).** The random sequence computed by the stochastic version of IADB presented in Algorithm 1 is a Markov chain. This algorithm might thus be reminiscent of stochastic diffusion models Song et al. (2021b) based on SDEs. However, it is not related to an SDE. Indeed, SDEs model stochastic behaviors at the infinitesimal scale while our mapping is stochastic for discrete steps and becomes a deterministic ODE in the infinitesimal limit.

**Non-Gaussian denoising diffusion.** Some previous works are dedicated to replacing Gaussian noise by other noise distributions such as the generalised normal (exponential power) distribution (Deasy et al., 2021) or the Gamma distribution (Nachmani et al., 2021). Our more general derivation works with any finite-variance density rather than specific noise alternatives. Peluchetti (Peluchetti, 2022) proposes a more general SDE framework. Our ODE can be derived from his SDE by nullifying the stochastic component and following the aggregation method.

**Alternative deterministic diffusion.** *Cold diffusion* (Bansal et al., 2022) shows that diffusion-like methods can reverse a variety of empirical image degradation processes. Our method is similar to their *animorphosis* example where human faces are progressively blended and deblended with animal faces. Our model provides a proven way to sample from the right density for this application.

**Image-to-image translation.** In Section 6, we have seen that IADB's mapping does not provide a faithful image-to-image translation. Previous works show that conditioning the diffusion process seem to be necessary to get faithful translations. For instance, adding an energy guide during the ODE integration (Zhao et al., 2022), by progressively injecting features (Meng et al., 2021), or by sampling conditional densities Saharia et al. (2021; 2022a). In Figure 10, we experiment with the latter with Gaussian noise for $x_0$, clean images for $x_1$, and a corrupted version of $x_1$ for the condition $c$ passed as an additional argument to the neural network: $D_\theta((1 - \alpha) x_0 + \alpha x_1, c, \alpha) = \bar{x}_1 - \bar{x}_0$.

condition ⊢—————generations—————     condition ⊢—————generations—————

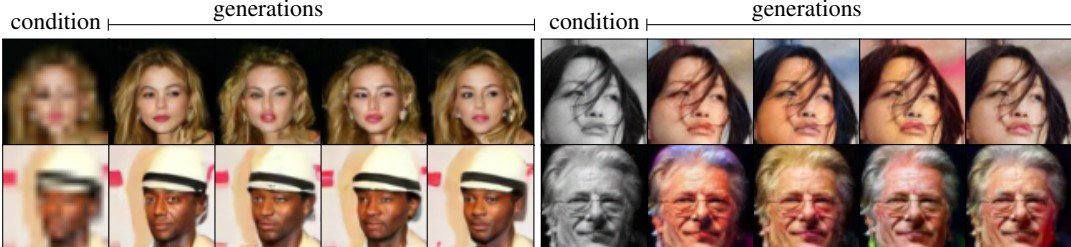

Figure 10: **Conditional image restoration with IADB.** From a corrupt image, either downscaling (left) or decolorization (right), we create various restorations using Gaussian noises $x_0$.

## 8 CONCLUSION

The objective of this work was to find a simple and intuitive way to approach deterministic diffusion. We derived Iterative $\alpha$-(de)Blending (IADB), a deterministic diffusion model based on a sampling interpretation of blending and deblending. Our model is similar to DDIM Song et al. (2021a) but its derivation is significantly simpler and reveals that the model is valid for arbitrary (finite-variance) densities rather than only Gaussian densities. Furthermore, our derivation leads to a variant learning formulation that happens to be more numerically stable than the one of DDIM.

An important takeaway of our experiments is that our results in image generation are significantly worse when using non-Gaussian densities. The theory allows to use non-Gaussian densities for sampling and our experiments on 2D non-Gaussian densities were successful in this regard. However, this did not transpose well to image generation in practice. This might be because we used neural networks that have been designed specifically for the denoising task or because denoising noise might be fundamentally simpler than deblending arbitrary images.

Finally, note that we experimented with our model in its vanilla setting with a uniform blending schedule and a first-order ODE solver. It might benefit from the improvements brought to denoising diffusion such as better blending schedules and higher-order ODE solvers (Karras et al., 2022).

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

# A LAW OF TOTAL PROBABILITY

The law of total probability states that the prior density $p_0 \times p_1$ can be written as a weighted linear combination of its posterior densities:

$$p_0(x_0)\, p_1(x_1) = \int_{\mathbb{R}^d} (p_0(x_0)\, p_1(x_1))_{|(x_\alpha, \alpha)}\; p_\alpha(x_\alpha)\, \mathrm{d}x_\alpha, \tag{6}$$

where $p_\alpha(x_\alpha)$ is the weight of each posterior density $(p_0(x_0)\, p_1(x_1))_{|(x_\alpha, \alpha)}$. As a result, sampling the prior density $p_0 \times p_1$ can be achieved by choosing a random posterior density proportionally to its weight, *i.e.* sampling $x_\alpha \sim p_\alpha$, and choosing a random sample in the posterior density conditioned by parameter $x_\alpha$, *i.e.* sampling $(x_0, x_1)_{|(x_\alpha \sim p_\alpha, \alpha)}$.

This is illustrated in Figure 11.

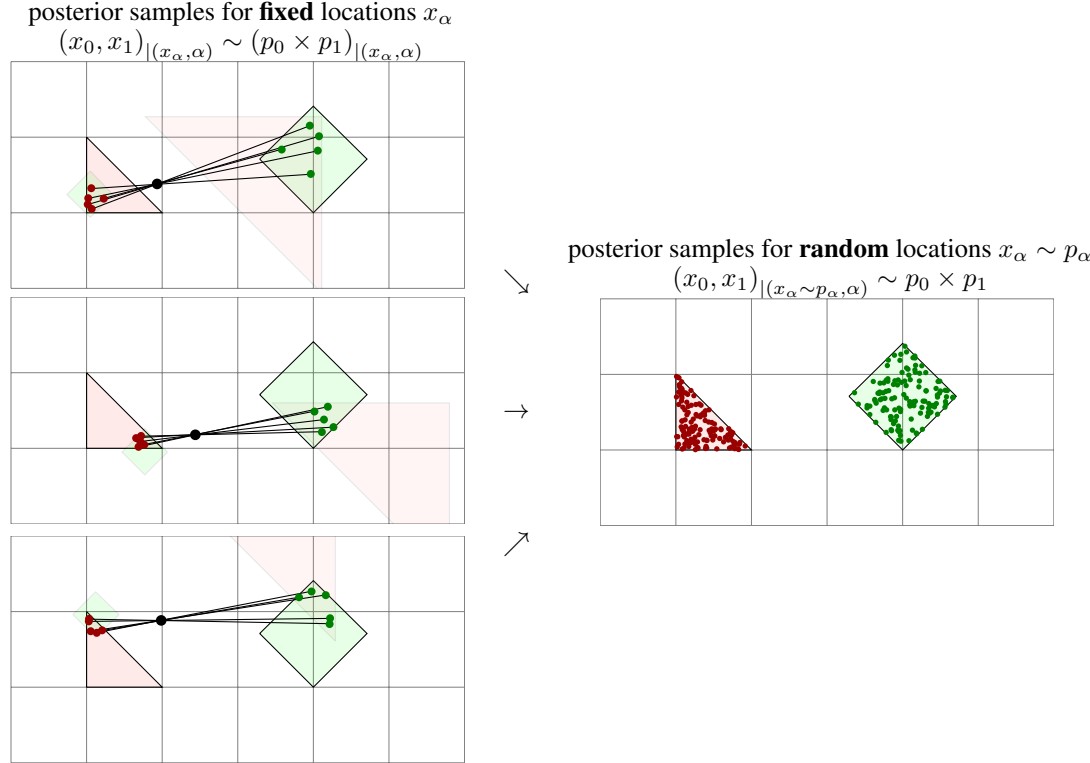

Figure 11: Each posterior density represents only a subset of the prior density. However, thanks to the law of total probability, we know that the average of random posterior densities is exactly the prior density. As a result, sampling random blended samples $x_\alpha \sim p_\alpha$ and random posterior samples $(x_0, x_1)_{|(x_\alpha, \alpha)} \sim (p_0 \times p_1)_{|(x_\alpha, \alpha)}$ is equivalent to sampling random $(x_0, x_1) \sim p_0 \times p_1$ directly in the prior density.

# B PROOF OF THE ITERATIVE $\alpha$-(DE)BLENDING CONVERGENCE THEOREM

## B.1 PRELIMINARIES

We first recall some properties that are required in the derivation of the limit of Algorithm 1.

**The posterior distributions have finite variance.** The theorem requires that $p_0$ and $p_1$ are of finite variance, such that their respective posterior densities $p_{0|(x,\alpha)}$ and $p_{1|(x,\alpha)}$ are also of finite variance. This is because the idea of the proof is that averaging many small random steps (provided by the posterior distributions) converges towards their expectations and it is true only if their variance is finite. We use this between Equation (23) and Equation (24).

**The expectations of the posterior distributions are continuous.** If $p_0$ and $p_1$ are classic, Riemann-integrable, densities, then they are continuous almost everywhere. Since the blended distributions are essentially convolutions of $p_0$ and $p_1$, it follows that the posterior densities $p_{0|(x,\alpha)}$ and $p_{1|(x,\alpha)}$ are also continuous almost everywhere and the expectation of their samples $x_{0|(x,\alpha)} \sim p_{0|(x,\alpha)}$ and $x_{1|(x,\alpha)} \sim p_{1|(x,\alpha)}$ are continuous everywhere (the expectation cancels out the null set where they are not continuous). In summary, for any $x \in \mathbb{R}^d$ and $\alpha \in [0,1]$ we have:

$$\lim_{x' \to x} \mathbb{E}\left[x_{0|(x',\alpha)}\right] = \mathbb{E}\left[x_{0|(x,\alpha)}\right], \quad \lim_{x' \to x} \mathbb{E}\left[x_{1|(x',\alpha)}\right] = \mathbb{E}\left[x_{1|(x,\alpha)}\right], \quad (7)$$

$$\lim_{\alpha' \to \alpha} \mathbb{E}\left[x_{0|(x,\alpha')}\right] = \mathbb{E}\left[x_{0|(x,\alpha)}\right], \quad \lim_{\alpha' \to \alpha} \mathbb{E}\left[x_{1|(x,\alpha)}\right] = \mathbb{E}\left[x_{1|(x,\alpha)}\right]. \quad (8)$$

We use this between Equation (24) and Equation (25).

## B.2 OBJECTIVE OF THE PROOF

To prove that Algorithm 1 and Algorithm 2 converge towards the same limit as the number of steps $T$ increases, we need to show that the trajectories of the samples are the same. This is the case if, in the limit, the derivatives $\frac{\mathrm{d}x_\alpha}{\mathrm{d}\alpha}$ are the same with both algorithms. The discrete update at step $t$ is:

$$\Delta\alpha_t = \alpha_{t+1} - \alpha_t = \frac{1}{T}, \quad (9)$$

$$\Delta x_{\alpha_t} = x_{\alpha_{t+1}} - x_{\alpha_t}, \quad (10)$$

and we want to prove that for any $\alpha \in [0,1]$ and at point $x_\alpha \in \mathbb{R}^d$ the continuous limit exists and is the same with both algorithms:

$$\frac{\mathrm{d}x_\alpha}{\mathrm{d}\alpha} = \lim_{\Delta\alpha \to 0} \frac{\Delta x_\alpha}{\Delta\alpha}. \quad (11)$$

## B.3 LIMIT OF ALGORITHM 2.

In step $t$ of Algorithm 2 we use the average of the posterior samples that are such that:

$$x_{\alpha_t} = (1 - \alpha_t)\,\bar{x}_{0|(x_{\alpha_t},\alpha_t)} + \alpha_t\,\bar{x}_{1|(x_{\alpha_t},\alpha_t)}, \quad (12)$$

$$x_{\alpha_{t+1}} = (1 - \alpha_{t+1})\,\bar{x}_{0|(x_{\alpha_t},\alpha_t)} + \alpha_{t+1}\,\bar{x}_{1|(x_{\alpha_t},\alpha_t)}, \quad (13)$$

where Equation (12) is a property of the average posteriors of $x_{\alpha_t}$ and Equation (13) is true by definition in Algorithm 2. We thus have the discrete difference:

$$\Delta x_{\alpha_t} = x_{\alpha_{t+1}} - x_{\alpha_t} = \Delta\alpha_t\left(\bar{x}_{1|(x_{\alpha_t},\alpha_t)} - \bar{x}_{0|(x_{\alpha_t},\alpha_t)}\right). \quad (14)$$

We obtain the discrete ratio

$$\frac{\Delta x_\alpha}{\Delta\alpha} = \bar{x}_{1|(x_\alpha,\alpha)} - \bar{x}_{0|(x_\alpha,\alpha)}, \quad (15)$$

which is independent of $\Delta\alpha$. The limit hence exists and is defined by

$$\boxed{\frac{\mathrm{d}x_\alpha}{\mathrm{d}\alpha} = \lim_{\Delta\alpha \to 0} \frac{\Delta x_\alpha}{\Delta\alpha} = \frac{\Delta x_\alpha}{\Delta\alpha} = \bar{x}_{1|(x_\alpha,\alpha)} - \bar{x}_{0|(x_\alpha,\alpha)}.} \quad (16)$$

### B.4 LIMIT OF ALGORITHM 1.

In step $t$ of Algorithm 1 we sample random posterior samples $x_{0|(x_{\alpha_t}, \alpha_t)}$ and $x_{1|(x_{\alpha_t}, \alpha_t)}$ that are such that:

$$x_{\alpha_t} = (1 - \alpha_t) \, x_{0|(x_{\alpha_t}, \alpha_t)} + \alpha_t \, x_{1|(x_{\alpha_t}, \alpha_t)}, \tag{17}$$

$$x_{\alpha_{t+1}} = (1 - \alpha_{t+1}) \, x_{0|(x_{\alpha_t}, \alpha_t)} + \alpha_{t+1} \, x_{1|(x_{\alpha_t}, \alpha_t)}, \tag{18}$$

where Equation (17) is a property of the posteriors of $x_{\alpha_t}$ and Equation (18) is true by definition in Algorithm 1. We thus have the discrete difference:

$$\Delta x_{\alpha_t} = x_{\alpha_{t+1}} - x_{\alpha_t} = \Delta \alpha_t \left( x_{1|(x_{\alpha_t}, \alpha_t)} - x_{0|(x_{\alpha_t}, \alpha_t)} \right). \tag{19}$$

We obtain the discrete difference for any parameter $\alpha \in [0, 1]$ and any location $x_\alpha \in \mathbb{R}^d$

$$\Delta x_\alpha = \Delta \alpha \left( x_{1|(x_\alpha, \alpha)} - x_{0|(x_\alpha, \alpha)} \right). \tag{20}$$

Furthermore, increasing the number of steps is equivalent to decomposing each step $\Delta \alpha$ into $N$ smaller steps $\Delta \alpha / N$. We rewrite the discrete difference as

$$\Delta x_\alpha = \frac{\Delta \alpha}{N} \sum_{n=0}^{N-1} \left( x_{1|(x_{\alpha + n\Delta\alpha/N}, \alpha + n\Delta\alpha/N)} - x_{0|(x_{\alpha + n\Delta\alpha/N}, \alpha + n\Delta\alpha/N)} \right). \tag{21}$$

With this modification, if the derivative exists, it is defined by the limit:

$$\frac{\mathrm{d}x_\alpha}{\mathrm{d}\alpha} = \lim_{\Delta\alpha \to 0} \lim_{N \to \infty} \frac{\Delta x_\alpha}{\Delta \alpha} \tag{22}$$

$$= \lim_{\Delta\alpha \to 0} \lim_{N \to \infty} \frac{1}{N} \sum_{n=0}^{N-1} \left( x_{1|(x_{\alpha + n\Delta\alpha/N}, \alpha + n\Delta\alpha/N)} - x_{0|(x_{\alpha + n\Delta\alpha/N}, \alpha + n\Delta\alpha/N)} \right). \tag{23}$$

Thanks to the **finite-variance** condition of $p_0$ and $p_1$, the normalized average sum converges towards the average of the posterior samples over $\alpha' \in [\alpha, \alpha + \Delta\alpha]$ as $N$ increases.

$$\frac{\mathrm{d}x_\alpha}{\mathrm{d}\alpha} = \lim_{\Delta\alpha \to 0} \mathbb{E}_{\alpha' \in [\alpha, \alpha + \Delta\alpha]} \left[ x_{1|(x_{\alpha'}, \alpha')} \right] - \mathbb{E}_{\alpha' \in [\alpha, \alpha + \Delta\alpha]} \left[ x_{0|(x_{\alpha'}, \alpha')} \right]. \tag{24}$$

Finally, because **the expectations of the posterior densities are continuous**, we obtain that the expectations over $[\alpha, \alpha + \Delta\alpha]$ converge towards the expectation in $\alpha$, such that

$$\boxed{\frac{\mathrm{d}x_\alpha}{\mathrm{d}\alpha} = \mathbb{E}\left[ x_{1|(x_\alpha, \alpha)} \right] - \mathbb{E}\left[ x_{0|(x_\alpha, \alpha)} \right] = \bar{x}_{1|(x_\alpha, \alpha)} - \bar{x}_{0|(x_\alpha, \alpha)}.} \tag{25}$$

This is the same result as in Equation (16) with Algorithm 2.

## C  VARIANT FORMULATIONS

We derive the variant formulations introduced in Section 3.1.

**Blended samples.**  A blended sample is by definition the blending of its posterior samples

$$x_{\alpha_t} = (1 - \alpha_t) \, x_0 + \alpha_t \, x_1. \tag{26}$$

Since blending is linear, a blended sample is also the blending of the average of its posterior samples:

$$x_{\alpha_t} = (1 - \alpha_t) \, \bar{x}_0 + \alpha_t \, \bar{x}_1. \tag{27}$$

We can thus rewrite its average posteriors samples $\bar{x}_0$ and $\bar{x}_1$ in the following way:

$$\bar{x}_0 = \frac{x_{\alpha_t}}{1 - \alpha_t} - \frac{\alpha_t \, \bar{x}_1}{1 - \alpha_t}, \tag{28}$$

$$\bar{x}_1 = \frac{x_{\alpha_t}}{\alpha_t} - \frac{(1 - \alpha_t) \, \bar{x}_0}{\alpha_t}. \tag{29}$$

**Variant (a):**  In the vanilla version of the algorithm, a blended sample of parameter $\alpha_{t+1}$ is obtained by blending $\bar{x}_0$ and $\bar{x}_1$:

$$x_{\alpha_{t+1}} = (1 - \alpha_{t+1}) \, \bar{x}_0 + \alpha_{t+1} \, \bar{x}_1. \tag{30}$$

**Variant (b):**  By expanding $\bar{x}_0$ from Equation (30) using Equation (28), we obtain:

$$x_{\alpha_{t+1}} = (1 - \alpha_{t+1}) \, \bar{x}_0 + \alpha_{t+1} \, \bar{x}_1, \tag{31}$$

$$= (1 - \alpha_{t+1}) \, \bar{x}_0 + \alpha_{t+1} \left( \frac{x_{\alpha_t}}{\alpha_t} - \frac{(1 - \alpha_t) \, \bar{x}_0}{\alpha_t} \right), \tag{32}$$

$$= \left( 1 - \alpha_{t+1} - \frac{\alpha_{t+1} \, (1 - \alpha)}{\alpha_t} \right) \bar{x}_0 + \frac{\alpha_{t+1}}{\alpha_t} \, x_{\alpha_t}, \tag{33}$$

$$= \left( 1 - \frac{\alpha_{t+1}}{\alpha_t} \right) \bar{x}_0 + \frac{\alpha_{t+1}}{\alpha_t} \, x_{\alpha_t}, \tag{34}$$

$$= \bar{x}_0 + \frac{\alpha_{t+1}}{\alpha_t} \, (x_{\alpha_t} - \bar{x}_0). \tag{35}$$

**Variant (c):**  By expanding $\bar{x}_1$ from Equation (30) using Equation (29), we obtain:

$$x_{\alpha_{t+1}} = (1 - \alpha_{t+1}) \, \bar{x}_0 + \alpha_{t+1} \, \bar{x}_1, \tag{36}$$

$$= (1 - \alpha_{t+1}) \left( \frac{x_{\alpha_t}}{1 - \alpha_t} - \frac{\alpha_t \, \bar{x}_1}{1 - \alpha_t} \right) + \alpha_{t+1} \, \bar{x}_1, \tag{37}$$

$$= \left( \alpha_{t+1} - \frac{(1 - \alpha_{t+1}) \, \alpha_t}{1 - \alpha_t} \right) \bar{x}_1 + \frac{1 - \alpha_{t+1}}{1 - \alpha_t} \, x_{\alpha_t}, \tag{38}$$

$$= \left( 1 - \frac{1 - \alpha_{t+1}}{1 - \alpha_t} \right) \bar{x}_1 + \frac{1 - \alpha_{t+1}}{1 - \alpha_t} \, x_{\alpha_t}, \tag{39}$$

$$= \bar{x}_1 + \frac{1 - \alpha_{t+1}}{1 - \alpha_t} \, (x_{\alpha_t} - \bar{x}_1). \tag{40}$$

**Variant (d):**  By rewriting $\alpha_{t+1} = \alpha_{t+1} + \alpha_t - \alpha_t$ in the definition of $x_{\alpha_{t+1}}$, we obtain:

$$x_{\alpha_{t+1}} = (1 - \alpha_{t+1}) \, \bar{x}_0 + \alpha_{t+1} \, \bar{x}_1, \tag{41}$$

$$= (1 - \alpha_{t+1} + \alpha_t - \alpha_t) \, \bar{x}_0 + (\alpha_{t+1} + \alpha_t - \alpha_t) \, \bar{x}_1, \tag{42}$$

$$= (1 - \alpha_t) \, \bar{x}_0 + \alpha_t \, \bar{x}_1 + (\alpha_{t+1} - \alpha_t) \, (\bar{x}_1 - \bar{x}_0), \tag{43}$$

$$= x_{\alpha_t} + (\alpha_{t+1} - \alpha_t) \, (\bar{x}_1 - \bar{x}_0). \tag{44}$$

## D  RELATION TO DDIM

In this section, we follow the notation of Song et al. (2021a): $x_0$ is a sample of a target density and $\epsilon$ is a random Gaussian sample. The denoiser of DDIM is defined such that, for an input $x_t = \sqrt{\alpha_t}x_0 + \sqrt{1-\alpha_t}\epsilon$, it learns

$$\epsilon^{(t)}(x_t) = \bar{\epsilon}. \tag{45}$$

We define

$$y_t = \frac{x_t}{\sqrt{\alpha_t} + \sqrt{1-\alpha_t}} \tag{46}$$

$$= \beta_t x_0 + (1-\beta_t)\epsilon, \tag{47}$$

with $\beta_t = \dfrac{\sqrt{\alpha_t}}{\sqrt{\alpha_t} + \sqrt{1-\alpha_t}}$. $y_t$ is an alpha-blended sample such as the one we defined in Section 2.
It follows that we have:

$$\frac{x_t}{\sqrt{\alpha_t}} = \frac{y_t}{\beta_t}. \tag{48}$$

We now turn to Equation (13) of Song et al. (2021a):

$$\frac{x_{t+1}}{\sqrt{\alpha_{t+1}}} = \frac{x_t}{\sqrt{\alpha_t}} + \left(\sqrt{\frac{1-\alpha_{t+1}}{\alpha_{t+1}}} - \sqrt{\frac{1-\alpha_t}{\alpha_t}}\right)\epsilon^{(t)}(x_t), \tag{49}$$

By injecting into this expression of the scaled coordinate at line 48, we obtain:

$$\frac{y_{t+1}}{\beta_{t+1}} = \frac{y_t}{\beta_t} + \left(\sqrt{\frac{1-\alpha_{t+1}}{\alpha_{t+1}}} - \sqrt{\frac{1-\alpha_t}{\alpha_t}}\right)\bar{\epsilon}, \tag{50}$$

and

$$y_{t+1} = y_t\frac{\beta_{t+1}}{\beta_t} + \beta_{t+1}\left(\sqrt{\frac{1-\alpha_{t+1}}{\alpha_{t+1}}} - \sqrt{\frac{1-\alpha_t}{\alpha_t}}\right)\bar{\epsilon}, \tag{51}$$

$$= y_t\frac{\beta_{t+1}}{\beta_t} + \frac{1}{\beta_t}\beta_t\beta_{t+1}\left(\sqrt{\frac{1-\alpha_{t+1}}{\alpha_{t+1}}} - \sqrt{\frac{1-\alpha_t}{\alpha_t}}\right)\bar{\epsilon}, \tag{52}$$

since

$$\beta_{t+1}\beta_t\left(\sqrt{\frac{1-\alpha_{t+1}}{\alpha_{t+1}}} - \sqrt{\frac{1-\alpha_t}{\alpha_t}}\right) = \beta_t(1-\beta_{t+1}) - (1-\beta_t)\beta_{t+1}, \tag{53}$$

$$= \beta_t - \beta_{t+1} \tag{54}$$

we can simply line 52 to:

$$y_{t+1} = y_t\frac{\beta_{t+1}}{\beta_t} + \frac{\beta_t - \beta_{t+1}}{\beta_t}, \bar{\epsilon} \tag{55}$$

$$= \bar{\epsilon} + y_t\frac{\beta_{t+1}}{\beta_t} - \frac{\beta_{t+1}}{\beta_t}\bar{\epsilon}, \tag{56}$$

$$= \bar{\epsilon} + \frac{\beta_{t+1}}{\beta_t}(y_t - \bar{\epsilon}). \tag{57}$$

This last form is exactly variant-(b) of IADB (see Table 1). To validate this claim, we trained a DDIM denoiser and applied the rescaling formula to the output samples. We show in Figure 12 that when we rescale the output of DDIM, the generated trajectory maps with IADB.

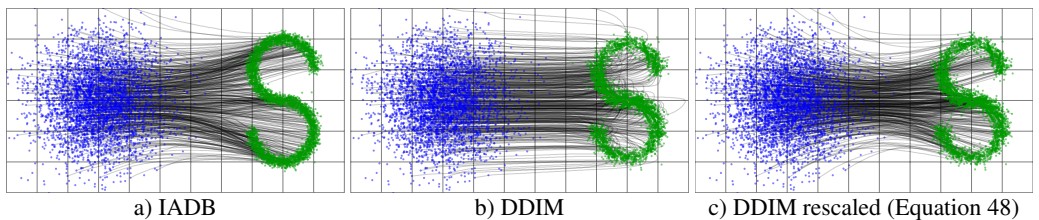

a) IADB         b) DDIM         c) DDIM rescaled (Equation 48)

Figure 12: We trained a MLP with 5 hidden layers of 64 filters to learn $D_\theta$ for IADB (a) and the same architecture to learn $\epsilon_\theta$ for DDIM (b) and (c). For (c), we convert points generated by DDIM using the scaling equation. The trajectories of the samples for IADB (a) and DDIM rescaled (c) match.

