# OpenReview forum: "Iterative $\alpha$-(de)Blending: Learning a Deterministic Mapping Between Arbitrary Densities"
_ICLR.cc/2023/Conference — Submitted to ICLR 2023_

### Official Review · Reviewer_XE6Y · 2022-10-18

**Confidence:** 4
**Correctness:** 3
**Technical Novelty And Significance:** 4
**Empirical Novelty And Significance:** 3
**Recommendation:** 6

**Clarity, Quality, Novelty And Reproducibility:**

**Clarity**. The paper is clear, nice to read, and highlights necessary takeaways while hiding away the details in the appendix.
I have spotted a couple of minor typos:
  - p. 2: "the have outperformed"
  - p. 5: "residual learning errors accumulates"
  - p. 6: "makes the neural network approximates"

**Quality**. The research is sound. The authors give necessary background information, and provide proofs of their claims, both theoretical and empirical. See above re: performance on non-Gaussian source densities.

I have a question though regarding the proof of the convergence theorem. It is stated that the finite variance is required. However, the proof seems to work even without this condition? Because the empiricial mean should converge to the expected value anyway. Please clarify if this is false.

**Novelty**. As to my knowledge, the result is novel.

**Reproducibility**. The authors do not provide source code but the experiment details are presented in the paper so should be easy to reproduce.


**Strength And Weaknesses:**

**Strengths**.
The paper clearly proposes a novel method to map densities. The theoretical background is sound and rigorous. The authors also support their claims by the empirical evidence. As the new method provides a significant improvement against the baseline, this is a major contribution to the field.

**Weaknesses**
Even though theoretically the approach could work with any densities, it turns out only Gaussian source density work well in practice. The authors give some alleged reasons for this behaviour but this could have been investigated a little more. Perhaps this is a direction for future work.


**Summary Of The Paper:**

The paper proposes an algorithm to determine a mapping between arbitrary densities. The algorithm is particularly conceptually simple compared to e.g. Langevin dynamics. The authors illustrate the algorithm on a variety of examples.


**Summary Of The Review:**

To sum up, the paper proposes a new method to map between densities. The method works well in Gaussian-to-arbritrary density setting. The paper is clearly written. Overall, my recommendation is accept.

UPDATE: in light of the other reviews, and in particular that the discussed paper can be seen as a a special case of the previous work [1], I would like to lower my score. This and the weaknesses I highlighted previously outweigh the strong points of the paper.

[1] Peluchetti, Stefano. "Non-Denoising Forward-Time Diffusions." (2021).

---

> ### Author Response · Authors · 2022-11-07
> **Answer to Reviewer XE6Y**
>
> Thank you for your very positive review and for pointing out the typos. We will fix them in our revision.
>
> > I have a question though regarding the proof of the convergence theorem. It is stated that the finite variance is required. However, the proof seems to work even without this condition? Because the empiricial mean should converge to the expected value anyway. Please clarify if this is false
>
> The empirical mean does not converge if the variance is not finite. For instance, if we consider i.i.d. variables $X_1, ... X_N$ such that var$(X_n) = \infty$, the variance of the empirical mean is var$(\frac{1}{N}\sum_{n=1}^N X_n) = \frac{1}{N^2}\sum_{n=1}^N$ var($X_n$) = $\frac{\infty}{N} = \infty$. This is the reason why we need the finite-variance condition in our theorem. To confirm experimentally that this condition is indeed necessary, we made an experiment with 2D densities such as the one of section 4 where we used a 2D Cauchy distribution for $f_0$ (Cauchy has infinite variance) and we observed empirically that the mapping never converged towards a deterministic mapping even with high number of samples.

---

### Official Review · Reviewer_j8qM · 2022-10-19

**Confidence:** 4
**Correctness:** 4
**Technical Novelty And Significance:** 4
**Empirical Novelty And Significance:** Not applicable
**Recommendation:** 6

**Clarity, Quality, Novelty And Reproducibility:**

The proposed deterministic denoising diffusion between arbitrary densities and the alternative more numerically stable learning formulation for DDIM appear novel to me. The approach is relatively simple, and the paper is easy to read and understand. The illustrations in Figure 4, 5, 6 are very helpful in understanding the dynamics of the proposed method.

One question: in Section 6, it is argued that "we regularize $f_0$ by applying a little amount of noise to the images. In theory, with this regularization, IADB is proven to produce a correct sampling of $f_1$ regardless of $f_0$". But in the Theorem on Page 4 which validates the proposed algorithm, it requires both $f_0$ and $f_1$ to be Riemann-integrable.

**Strength And Weaknesses:**

Strengths:
- The proposed iterative $\alpha$-(de)blending (IADB) can be considered as a more general denoising diffusion process that can (in theory) work with arbitrary densities, which has the DDIM as one of special cases when the target density is Gaussian. The main idea is simple, and the nice thing about IADB is that the derivation does not rely on complex concepts which makes it more flexible. I really like the underlying motivation of the work to find a simpler approach to deterministic diffusion using only basic sampling concepts.
- With the flexibility of the formulation, the authors are able to derive different learning variants where one of them is more numerically stable than others. The results on LSUN Bedrooms, CelebA and AFHQ Cats are favorable comparing to DDIM in terms of FID.

Weakness:
- The proposition on Page 3 and the proof in Appendix A could be improved. This proposition is a critical part in understanding the proposed method and the formula used for Bayes' theorem is not the famous one ($P(A |B) = P(B | A) P(A) / P(B)$), thus it deserves more discussion in the main text.
- The experimental comparison between the proposed IADB and DDIM. It is nice that the authors try to control all the conditions so that the performance difference can be solely attributed to the different learning processes. But the resulting FID scores for the DDIM are considerably worse than those reported in the original paper, this may raise questions about the validity of the comparison.
- The disappointing results with arbitrary image densities. As one of the main contributions of this work, neither the image generation nor restoration gives satisfactory results among arbitrary image densities. The authors argue that "experimental set up that works well with Gaussian noise does not necessarily transpose successfully to other densities". It would be much better if the authors could provide an architectural design choice that works.

**Summary Of The Paper:**

The goal of this paper is to develop a deterministic denoising diffusion between arbitrary densities without relying on complex concepts like Langevin dynamics or score-matching. Thanks to Bayes' theorem, it shows that iteratively blending and deblending samples produce random paths between arbitrary densities and proves this converges to a deterministic mapping that can be learned to deblend samples. Here, blending means generating samples from a convex combination of the start and end densities, while deblending means generating samples from the posterior distributions of the start and end densities given the blended distribution and the blending parameter.

The equivalence to the DDIM in a special situation has been established and a more numerically stable sampling process is proposed. The main evaluations on image generation with Gaussian noise as the target density indeed demonstrate a favorable sample quality according to FID. On the other hand, disappointing generation results are also illustrated between different image datasets.

**Summary Of The Review:**

In summary, the paper provides a simpler but more flexible derivation of deterministic denoising diffusion models that works with arbitrary start and end densities and has the DDIM as a special case when the target density is Gaussian. The flexibility of the formulation also warrants different learning variants where one of them is more numerically stable than others and compares favorably with DDIM in terms of image generation quality measured by FID. Despite the disappointing results with arbitrary image densities, I think this work is a nice addition to the community.

---

> ### Author Response · Authors · 2022-11-07
> **Answer to Reviewer j8qM**
>
> > The proposition on Page 3 and the proof in Appendix A could be improved. This proposition is a critical part in understanding the proposed method and the formula used for Bayes' theorem is not the famous one ($P(A|B) = P(B|A) P(A) / P(B)$), thus it deserves more discussion in the main text.
>
> You are right, we are not exactly using Bayes' theorem. Thank you for pointing it out. What we use more specifically is the Law of Total Probability, which underlies Bayes' theorem: https://en.wikipedia.org/wiki/Law_of_total_probability
> This theorem is usually used to prove Bayes' one. We will replace Bayes' theorem by this one in our revision.
>
> > The experimental comparison between the proposed IADB and DDIM. It is nice that the authors try to control all the conditions so that the performance difference can be solely attributed to the different learning processes. But the resulting FID scores for the DDIM are considerably worse than those reported in the original paper, this may raise questions about the validity of the comparison.
>
> This discrepancy comes from the number of images generated to compute the FID scores. Generating images with 128 steps takes a huge amount of times and in the submitted version of our paper we used only 5k images per dataset, which is just above the minimal recommended amount. In the meanwhile, we tested with 50k images, as recommended in the original FID paper, and noticed that the scores are much smaller. For instance, with 50k images with 100 steps on the CelebA dataset we obtain FID scores of 6.9516 for DDIM and 5.7278 for IADB. This is in accordance with the scores presented in the DDIM paper. We will update all the FID numbers in our revision with 50k images. Note that this does not change the outcome of the experiment: IADB is consistently better than DDIM with 50k-image FID scores.
>
> > The disappointing results with arbitrary image densities. As one of the main contributions of this work, neither the image generation nor restoration gives satisfactory results among arbitrary image densities. The authors argue that "experimental set up that works well with Gaussian noise does not necessarily transpose successfully to other densities". It would be much better if the authors could provide an architectural design choice that works.
>
> We wish to better frame how to analyze our results in our section 6.
>
> In the experiment of Figure 8, we note that the quality is not on par with SOTA Gaussian-noise architectures. Those have been carefully tuned over multiple papers, authors, and Github contributors to provide high quality with Gaussian noise. We wish we could provide an architecture that would be designed specifically for non-Gaussian inputs and achieve SOTA quality but this is an orthogonal problem to the theoretical formulation. Here we wanted to contribute on the theoretical derivation of deterministic diffusion models and hopefully it will inspire more research on the practical architecture designs.
>
> In the experiment of Figure 9, we observe that the mapping does not act as a restoration process. This is independent of the architecture design, this is just how the mapping behaves. Remember that, besides the parameterization and the learning objective, our mapping is the same as DDIM (or other works as pointed out by Reviewer rt47). It is thus a pretty legitimate question to check whether deterministic denoising diffusion can act as an image restoration process with unpaired data. We experimented with this and report in Figure 9 that this is not the case. If other people used DDIM or another equivalent deterministic diffusion model with non-Gaussian inputs and make the same experiment, they would reach the same conclusion. We believe that this is knowledge worth sharing about the behaviour of this family of ODEs.
>
> > In Section 6, it is argued that "we regularize  by applying a little amount of noise to the images. In theory, with this regularization, IADB is proven to produce a correct sampling of $f_1$ regardless of $f_0$". But in the Theorem on Page 4 which validates the proposed algorithm, it requires both $f_0$ and $f_1$ to be Riemann-integrable.
>
> Adding a small amount of noise to the samples of a distribution means convolving the distribution with a Gaussian kernel, which makes it Riemann-integrable. Hence, by adding a small amount of noise to the data, we make sure that the theorem applies.

---

### Official Review · Reviewer_W2Pp · 2022-10-24

**Confidence:** 4
**Correctness:** 3
**Technical Novelty And Significance:** 2
**Empirical Novelty And Significance:** 2
**Recommendation:** 3

**Clarity, Quality, Novelty And Reproducibility:**

The paper is generally well-written. The idea is interesting, but its practical value is unclear, as the proposed method cannot obtain high quality samples with non-Gaussian densities.

**Strength And Weaknesses:**

### Strong points

- The paper proposed a generalized version of DDIM that allows for bridging arbitrary distribution. The idea itself is interesting and provides an alternative perspective to the well-known deterministic sampling process of diffusion models.

- The proposed training and sampling algorithms are straightforward to implement.

- Experimentally, their method outperforms DDIM on several image datasets when starting from Gaussian, and they showcase the generalization to arbitrary pairs of distributions.

### Weak points

- The paper lacks some important baselines that can also bridge two arbitrary distributions. For example, people can use the bijection map in [1] to first transform distribution A to a distribution on a large hemisphere by the forward-process, and then transform the distribution on hemisphere back to distribution B by a learned backward-process. The simple "bridging" process can also define a bijection between an arbitrary pair of distributions.

- The experiments in this paper do not back up the paper’s claim. In particular, the main claim of this paper is that the proposed method can generalize to a generative process between two arbitrary distributions (Section 6). However, also as the authors pointed out, in practice, the performances on these tasks are disappointing, and the positive results only come when fixing the prior sampling distribution to be a Gaussian one. The reviewer has one possible hypothesis: it's difficult for neural networks to learn the difference between two heterogeneous images, such as Leave and CelebA. Instead, the original approach of diffusion models (variant (b) in Table 1) would be better. Is it possible to verify the hypothesis and give the rationale behind the failure?

- The dataset of pure image generation in Section 5 is easy -- they are all single-mode datasets. Could you provide experiments on more challenging datasets, such as CIFAR-10? In addition, it seems that the only difference between the proposed method and DDIM is the network target (or parameterization of the score function). Could you elaborate more on this part of the paper?

- The second experiment in Section 6 is effectively an inverse problem. The author could resort to some manifold-constraint methods such as [2] to see if they help.

### Writing suggestions / Minors

- All theorems and propositions should be numbered.
- The reviewer is confused about the listed baselines of DDIM. When sampling 128 steps, DDIM is shown to give 18.70 on CelebA $64^2$, while in the original paper, it is 6.53 with 100 steps. I am wondering why there is such a big discrepancy? If the reason is that the models have not yet been trained to convergence, I suggest the authors report the comparison at convergence.

- The reviewer suggests changing the density notation from $f$ to $p$, as $p$ is more commonly used in the community.

- In Section 1: (1) SDE (such as reverse diffusion) is not a variant of Langevin dynamics but a complementary method. (2) " We derive a deterministic diffusion model ..." The authors' model is more general than the "deterministic diffusion model".

[1] Poisson Flow Generative Models, NeurIPS 2022, https://arxiv.org/abs/2209.11178

[2] Improving Diffusion Models for Inverse Problems using Manifold Constraints, NeurIPS 2022, https://arxiv.org/abs/2206.00941


**Summary Of The Paper:**

The paper proposes a new (bijective) transformation method between arbitrary pairs of distributions. The idea is to iteratively blend and de-blend the samples along the trajectories. Effectively, the direction in each iteration is the difference between the mean of the posterior samples in the two distributions. The method can be viewed as a generalization of DDIM, a deterministic sampling method for diffusion models. If the starting distribution is standard normal, they are equivalent up to the network parameterization. Experimentally, their method successfully transforms from one distribution to the other.

**Summary Of The Review:**

Given the lack of basic baselines and the unclear practical value, the reviewer leans toward rejection for current version.

---

> ### Author Response · Authors · 2022-11-07
> **Answer to Reviewer W2Pp (1/2)**
>
> Thank you for the writing suggestions and the minor comments. We will fix them in our revision.
>
> > The idea is interesting, but its practical value is unclear, as the proposed method cannot obtain high quality samples with non-Gaussian densities.
>
> Before answering specific questions, we wish to point out that there seem to be some misunderstanding between the paper's objective and this review. This review rates the paper only on practical considerations. However, as stated in our  abstract and introduction, the paper's main objective is to provide a novel view and a simpler entry point in deterministic denoising diffusion. Indeed, we found it hard to start learning about this topic. We had to understand Langevin's diffusion, SDEs, score matching, etc. before getting to the foundation of DDIM. This is a lot of advanced math for something that turns out to be pretty simple in terms of implementation. Our philosophy is that simple things deserve simple explanations. So, we started looking for a derivation that could produce a similar mapping but based only on simple concepts. This is what the paper proposes. Unfortunately, this objective is not acknowledged at all in the review.
>
> On the practical side, there are some interesting bonus properties, such that the mapping works for non-Gaussian $f_0$ densities, and that the formulation performs slightly better than DDIM with Gaussian $f_0$. But to us, they are just nice side effects. They are not the paper's main objective. The review is sadly only focused on those.

---

> > ### Author Response · Authors · 2022-11-07
> > **Answer to Reviewer W2Pp (2/2)**
> >
> > > The paper lacks some important baselines that can also bridge two arbitrary distributions. For example, people can use the bijection map in [1] to first transform distribution A to a distribution on a large hemisphere by the forward-process, and then transform the distribution on hemisphere back to distribution B by a learned backward-process. The simple "bridging" process can also define a bijection between an arbitrary pair of distributions.
> >
> > It is true that an arbitrary mapping can be built in this indirect way. However, we did not find experimental evidences in the literature that this would work in practice. We will still mention as a theoretical possibility. Another difference is that, in our case, the mapping is learnt directly while the *bridge* approach would require to train two networks separately.
> >
> > > The experiments in this paper do not back up the paper’s claim. In particular, the main claim of this paper is that the proposed method can generalize to a generative process between two arbitrary distributions (Section 6). However, also as the authors pointed out, in practice, the performances on these tasks are disappointing, and the positive results only come when fixing the prior sampling distribution to be a Gaussian one. The reviewer has one possible hypothesis: it's difficult for neural networks to learn the difference between two heterogeneous images, such as Leave and CelebA. Instead, the original approach of diffusion models (variant (b) in Table 1) would be better. Is it possible to verify the hypothesis and give the rationale behind the failure?
> >
> > We disagree with the fact that the paper's claim are not backed up experimentally. Our 2D experiments in Section 4 are successful and our image experiments are not a failure. As a matter of fact, our model indeed maps
> >  - Pebble to Celeba
> >  - Leave to Celeba
> >  - Cat to Dog
> >  - Map to Sat
> >
> > In each experiment, the result is effectively a meaningful sample from the target density. It is just that the quality is not as good as with Gaussian noise. Not showing SOTA quality with a new model does not mean that the model is invalidated. For instance, diffusion models used to be inferior to GANs at first. Did it mean that the early diffusion experiments did not back up the theoretical claims? In our case, we used an architecture from the *Diffusers* framework that has been designed and refined for Gaussian noise across a wide range of papers, authors, and Github contributors. We can hardly compete with them by starting from scratch in a single paper.
> >
> > > The dataset of pure image generation in Section 5 is easy -- they are all single-mode datasets. Could you provide experiments on more challenging datasets, such as CIFAR-10?
> >
> > We will add a CIFAR-10 comparison to DDIM in our revision.
> >
> > > In addition, it seems that the only difference between the proposed method and DDIM is the network target (or parameterization of the score function). Could you elaborate more on this part of the paper?
> >
> > In practice, the main differences with DDIM are the network learning objective (variant d versus variant b) and the parameterization and there is indeed a lot of similarity. As explained above, the main contribution of our paper is the derivation: it is simpler and requires less technical background than the one of DDIM, which is based on SDEs, and provides a more general result.
> >
> > > The reviewer is confused about the listed baselines of DDIM. When sampling 128 steps, DDIM is shown to give 18.70 on CelebA , while in the original paper, it is 6.53 with 100 steps. I am wondering why there is such a big discrepancy? If the reason is that the models have not yet been trained to convergence, I suggest the authors report the comparison at convergence.
> >
> > This discrepancy comes from the number of images generated to compute the FID scores. Generating images with 128 steps takes a huge amount of times  and in the submitted version of our paper we used only 5k images per dataset to compute the FID, which is just above the minimal recommended amount. In the meanwhile, we tested with 50k images, as recommended in the original FID paper, and noticed that the scores are much smaller. For instance, with 50k images with 100 steps on the Celeba dataset we obtain FID scores of 6.9516 for DDIM and 5.7278 for IADB. This is in accordance with the scores presented in the DDIM paper. We will update all the FID numbers in our revision with 50k images. Note that this does not change the outcome of the experiment: IADB is consistently better than DDIM with 50k-image FID scores.

---

> > ### Comment · Reviewer_W2Pp · 2022-11-13
> > **Response**
> >
> > Thank the authors for their response. After reading the response and re-accessing the paper, I decide to decrease the scores to 3.  I could reconsider raising the score if the authors at least address some of my concerns.
> >
> > ### 1. "This review rates the paper only on practical considerations. However, as stated in our abstract and introduction, the paper's main objective is to provide a novel view and a simpler entry point in deterministic denoising diffusion.   ... But to us, they are just nice side effects. They are not the paper's main objective. The review is sadly only focused on those."
> >
> > I do appreciate the novel angle provided in this paper. However, I disagree with the authors that it's a "simpler entry point". As a practitioner in diffusion models, I feel that the forward-backward SDE framework in [1, 2] is clearer than the blend/deblend perspective.  They provide a more unified and expansible framework, sparking lots of follow-up work. Indeed, their papers are more educational and instructive for practitioners. In addition, I wouldn't call those linear/semi-linear forward ODE/SDEs ``advanced math".
> >
> > On the other hand, empirical performance is a key metric for accessing generative models. The popularity of diffusion models/score-based models, e.g., stable diffusion, dream booth, diffdock, arose from their strong empirical performance and stability, not just the theory. The "side effects" statement from the author is very unfair, as lots of researchers/papers in this field take many efforts to build high-performing generative models, in addition to their nice theory.
> >
> > *[1] Score-Based Generative Modeling through Stochastic Differential Equations. Yang Song, Jascha Sohl-Dickstein, Diederik P. Kingma, Abhishek Kumar, Stefano Ermon,  and Ben Poole. The 9th International Conference on Learning Representations, 2021.*
> >
> > *[2] Elucidating the Design Space of Diffusion-Based Generative Models. Tero Karras, Miika Aittala, Timo Aila, Samuli Laine. https://arxiv.org/abs/2206.00364*
> >
> > ### 2. "It is true that an arbitrary mapping can be built in this indirect way. However, we did not find experimental evidences in the literature that this would work in practice. ...  Another difference is that, in our case, the mapping is learnt directly while the bridge approach would require to train two networks separately."
> >
> > I do find some papers that give strong empirical results in this indirect way through the Diffusion ODE/SDEs, such as [1] and their baselines. These approaches don't require training two networks in principle -- it can be achieved by domain-conditioning generation using a single network.
> >
> > *[1] Unifying Diffusion Models' Latent Space, with Applications to CycleDiffusion and Guidance. Chen Henry Wu, Fernando De la Torre. https://arxiv.org/abs/2210.05559*
> >
> >
> > ### 3. " We disagree with the fact that the paper's claim are not backed up experimentally. Our 2D experiments in Section 4 are successful and our image experiments are not a failure."
> >
> > The authors themselves mention that the results are disappointing. To see how to illustrate the effectiveness of the domain transfer quantitatively, please refer to the CycleDiffusion paper above.
> >
> >
> > ### 4. "Not showing SOTA quality with a new model does not mean that the model is invalidated. For instance, diffusion models used to be inferior to GANs at first. Did it mean that the early diffusion experiments did not back up the theoretical claims? "
> >
> > I disagree with the authors. Diffusion models showed large potential at first, for example, it doesn't have the adversarial training objective in GANs. Also, I would say GANs and diffusion models are two completely different models. I think the author only provides another (incremental) perspective for diffusion models.
> >
> > ### 5. "We can hardly compete with them by starting from scratch in a single paper."
> >
> > Again, papers [1, 2, 3] take many efforts to tune the architectures of their new models, in order to showcase the utility. I think the authors' statement is unfair to other papers. In addition, the blending methods on two discrete datasets (like domain transfer) defines deterministic paths between the domain: it's impossible for two pair of images to blend to the same intermediate point for finite datasets. In this case, it's hard for the neural model to capture the learning signal in the blending framework.
> >
> >
> > *[1] Score-Based Generative Modeling through Stochastic Differential Equations. Yang Song, Jascha Sohl-Dickstein, Diederik P. Kingma, Abhishek Kumar, Stefano Ermon,  and Ben Poole. The 9th International Conference on Learning Representations, 2021.*
> >
> > *[2] Elucidating the Design Space of Diffusion-Based Generative Models. Tero Karras, Miika Aittala, Timo Aila, Samuli Laine. https://arxiv.org/abs/2206.00364*
> >
> > *[3] Generative Modeling by Estimating Gradients of the Data Distribution. Yang Song,  and Stefano Ermon. The 33rd Conference on Neural Information Processing Systems, 2019.*

---

> > > ### Comment · Reviewer_W2Pp · 2022-11-13
> > > **Response (cont.)**
> > >
> > > ### 6. "We will add a CIFAR-10 comparison to DDIM in our revision."
> > >
> > > I'm wondering why the author couldn't provide the CIFAR-10 results during the rebuttal period, as its size is relatively small. CIFAR-10 is the most standard benchmark for accessing generative models. It's more challenging than the datasets used in this paper, as it has multiple classes.
> > >
> > > ### 7. "In practice, the main differences with DDIM are the network learning objective (variant d versus variant b) and the parameterization and there is indeed a lot of similarity. As explained above, the main contribution of our paper is the derivation: it is simpler and requires less technical background than the one of DDIM, which is based on SDEs, and provides a more general result."
> > >
> > > The network learning objective is the same as the network target (or parameterization of the score function) I mentioned in the original question. It's unclear to me why the new parameterization helps in unconditional image generation. For the argument of simplicity over SDEs, please see the first paragraph in point 1 above.

---

> > > > ### Author Response · Authors · 2022-11-18
> > > > **Answer**
> > > >
> > > > The previous exchange has spanned multiple points. There were two points that requested experiments: CIFAR-10 and image-to-image mappings.
> > > >
> > > > > I'm wondering why the author couldn't provide the CIFAR-10 results during the rebuttal period, as its size is relatively small.
> > > >
> > > > **CIFAR-10.** Despite CIFAR-10 images are pretty small, the experiment is not that fast to run. The authors of DDIM used the CIFAR-10 model trained by the authors of DDPM (from Google). The DDPM paper explains that they trained CIFAR-10 for 10h on a TPU v3-8 (similar to 8 V100 GPUs). The order of magnitude of equivalent training time on a customer GPU is several days. We hence only have unconverged results for now. Here are the FIDs that we currently have:
> > > >
> > > > |             | T=2           | T=4         | T=8          | T=16          | T=32          | T=128         |
> > > > | ----------- | ------------- | ----------- | ------------ | ------------- | ------------- | ------------- |
> > > > | DDIM        | 216.10        | **89.96**   | 61.87        | 52.98         | 27.45         | 28.42         |
> > > > | IADB        | **186.40**    | 90.98       | **48.72**    | **29.60**     | **22.24**     | **18.88**     |
> > > >
> > > > We will let them converge for the final version of the paper.
> > > >
> > > > *Update (08/12/2022) we now have the following scores:*
> > > > |             | T=2           | T=4         | T=8          | T=16          | T=32          | T=128         |
> > > > | ----------- | ------------- | ----------- | ------------ | ------------- | ------------- | ------------- |
> > > > | DDIM        | 215.86        | 82.42   | 39.48        | 23.93         | 15.96         | 9.70         |
> > > > | IADB        | **166.17**    | **62.17**       | **25.06**    | **14.26**     | **10.42**     | **8.15**     |
> > > >
> > > >
> > > > **Image-to-image mappings.** We did not make comparisons because we do not have a competitive claim for these applications. We clearly admit that the resulting mapping is not faithful in the sense of what a human user would expect (the theorem claims that there is a bijective mapping, not that it is humanly faithful). We do not see why it is a problem to report this. Reviewer rt47 pointed this as a strength of the paper. Furthermore, the outcome of our experiment is not that surprising after all. To force the mapping to be faithful, additional guidance is commonly added in previous works. For instance, [1] adds an energy guide during the ODE integration, [2] progressively injects features, and [3] uses a conditional formulation. These methods do not rely solely on the S/O-DE mapping to get something humanly faithful. Our conclusion is that the ODE mapping alone, without further guidance, is generally not enough for guaranteed faithful image-to-image applications. This is not in opposition with these previous works. We have updated our related work section to explain this.
> > > >
> > > > [1] Unpaired image-to-image translation via energy-guided stochastic differential equations. Min Zhao, Fan Bao, Chongxuan Li, and Jun Zhu.
> > > >
> > > > [2] Sdedit: Image synthesis and editing with stochastic differential equations. Chenlin Meng, Yang Song, Jiaming Song, Jiajun Wu, Jun-Yan Zhu, and Stefano Ermon.
> > > >
> > > > [3] Image super-resolution via iterative refinement. Chitwan Saharia, Jonathan Ho, William Chan, Tim Salimans, David J. Fleet, and Mohammad Norouzi.

---

> > > > > ### Comment · Reviewer_W2Pp · 2022-12-09
> > > > > **The view is not novel**
> > > > >
> > > > > Thank you for the response. The proposed IADB changes the noise schedule during training [1] per re-parameterization, and the target (now $x_1-x_0$). It's not entirely clear why such modifications would improve DDIM.
> > > > >
> > > > > I would like to point out that the view proposed in the paper is not novel (also mentioned by Reviewer rt47). A similar view has been developed in [2]. They consider a more sophisticated Brownian motion between two fixed points, instead of the linear interpolation in the current paper. The $\alpha$-blending framework can be deduced from their mixture argument. The linear case considered in the current paper can be regarded as a degenerated case of [2].
> > > > >
> > > > > Due to the limited theoretical contribution, and the unclear utility of the proposed algorithm (the authors didn't provide additional convincing experimental results beyond similar theoretical arguments in [2]) , I tend to keep my score.
> > > > >
> > > > >
> > > > >
> > > > > [1] Score-Based Generative Modeling through Stochastic Differential Equations. Yang Song, Jascha Sohl-Dickstein, Diederik P. Kingma, Abhishek Kumar, Stefano Ermon, and Ben Poole. The 9th International Conference on Learning Representations, 2021.
> > > > >
> > > > > [2] Non-Denoising Forward-Time Diffusions, https://openreview.net/forum?id=oVfIKuhqfC

---

### Official Review · Reviewer_rt47 · 2022-11-04

**Confidence:** 4
**Correctness:** 3
**Technical Novelty And Significance:** 3
**Empirical Novelty And Significance:** 3
**Recommendation:** 6

**Clarity, Quality, Novelty And Reproducibility:**

* Very clear writing. Theoretical analysis and empirical results are solid.
* Originality of this work is less ideal, as some major ideas in this work actually existed before, yet authors failed to cite and compare with them.

**Strength And Weaknesses:**

## Strength

1. The formulation is simple yet effective. Writing is clear and easy to understand.
2. The proposed method generalizes existing deterministic diffusion models (DDIM in particular), offering new insights into their inner workings.
3. Negative results on image translation and non-Gaussian diffusion are thought-provoking.

## Weaknesses

1. The proposed idea is nice but not entirely new. The formulation can be reduced to a special case of the method in [1]. Specifically, one can set the SDE to $d x_t = (x_1 - x_0) dt + 0 dw$ (which actually becomes an ODE) and leverage Theorem 1 in [1] to derive the aggregated  ODE $d x_t = E[x_1 - x_0 | x_t] dt$, where the conditional expectation can be estimated by training a neural network $D_\theta(x_t, t)$ with mean squared error minimization. This recovers the exact formulation in Algorithm 4, and yields the same exact training objective in equation (5) in this paper. Unfortunately, the work of [1] is not cited nor discussed in this paper.

2. The connection between $\alpha$-(de)blending and DDIM is implied in the work [2] (see Appendix D). Specifically, you can understand $x$, $\epsilon$, $z_\phi$ in [2] as $x_0$, $x_1$ and $x_\alpha$ in this work. The (d) formulation in Table 1 amounts to $v$-prediction in [2]. In light of this connection, it is important to discuss and credit the work of [2].

## References
[1] Peluchetti, Stefano. "Non-Denoising Forward-Time Diffusions." (2021).

[2] Salimans, Tim, and Jonathan Ho. "Progressive distillation for fast sampling of diffusion models." arXiv preprint arXiv:2202.00512 (2022).

**Summary Of The Paper:**

This paper proposes a formulation of discrete diffusion models that do not rely on the concept of score matching or stochastic differential equations. Instead, authors leverage an interpolation function for samples from two distributions, and show that an iterative interpolation ($\alpha$-blending) and inverse interpolation ($\alpha$-deblending) process can transport samples from one distribution to the other. Authors further provide training objectives for this process and prove that it is a generalization of conventional deterministic diffusion models, such as DDIM. Empirically, authors obtain better sample quality than DDIMs with fewer sampling iterations. Lastly, authors report experimental results of using non-Gaussian noise for sample generation.

**Summary Of The Review:**

The idea is simple, intuitive, yet very effective. Experiments provide insights into when the method works better, and when not. However, this work lacks discussion of some previous papers that explored closely related ideas.

---

> ### Author Response · Authors · 2022-11-07
> **Answer to Reviewer rt47**
>
> Thank you for bringing these two papers to our attention.
>
> > The proposed idea is nice but not entirely new. The formulation can be reduced to a special case of the method in [1]. Specifically, one can set the SDE to $dx_1 = (x_1-x_0) dt + 0dw$  (which actually becomes an ODE) and leverage Theorem 1 in [1] to derive the aggregated ODE $dx_t=\mathbb{E}[x_1-x_0 | x_t]dt$, where the conditional expectation can be estimated by training a neural network $D_\theta(x_t, t)$ with mean squared error minimization. This recovers the exact formulation in Algorithm 4, and yields the same exact training objective in equation (5) in this paper. Unfortunately, the work of [1] is not cited nor discussed in this paper.
>
> This is a paper that we missed. We note that it is a paper rejected from ICLR 2022 and that has not been published somewhere else, which is why it went under our radar. We agree that our ODE can be framed as a special case of their SDE and we will acknowledge it in our revision.
>
> Nonetheless, we do not believe that it lessens our contribution more than DDIM already does. Indeed, we already acknowledge that DDIM's ODE is the same as ours besides the parameterization and the variant (b versus d in our Table 1). It is the same with this paper. Furthermore, the situation in [1] is exactly what we described in our introduction: the reader is expected to master a significant amount of advanced knowledge about SDEs before getting to an ODE that turns out to be very simple. According to [1]'s review summary, one of the reason it was rejected is because "*several reviewers found the presentation confusing and overly complex, including the notation, writing, and figures*". The motivation for our paper is precisely to derive this ODE without any SDE background. Our main claim is thus not so much that this ODE is new but rather that we bring it to the reader in a simple and clear way.
>
> > The connection between $\alpha$-(de)blending and DDIM is implied in the work [2] (see Appendix D). Specifically, you can understand $x$, $\epsilon$, $z_\theta$ in [2] as $x_0$, $x_1$ and $x_\alpha$ in this work. The (d) formulation in Table 1 amounts to -prediction in [2]. In light of this connection, it is important to discuss and credit the work of [2].
>
> We were aware of the progressive-distillation paper but missed the connection with the different formulations in our Table 1. We acknowledge that the three formulations proposed in this paper are, besides the variable changes, equivalent to the ones we propose (learning the noise, the image, or the noise->image direction) and we are going to make the connection and credit them in our revision. Still, we believe that the way we frame the variants reveals the numerical stability of our formulation (d) that does not involve any division. In the distillation paper, three different learning objectives are exposed but they are all meant to be recombined into a $x$ used to evaluate the *same* stepping scheme, which involves a division similar to our variant (c). In contrast, our Table 1 provides 4 learning objectives and 4 corresponding stepping schemes, which reveals the stability issue.

---

### Author Response · Authors · 2022-11-18
**Revision upload**

We thank the reviewers for their feedback. We uploaded our revision with the following changes:

 - We recomputed all the FIDs with 50k images. They are now on par with the FIDs presented in the DDIM paper. The conclusion for our method remains the same.

 - We refer to the Law of total probability rather than Bayes' theorem.

 - We cite [Peluchetti2022] and credit that our ODE can be framed as a special case of his SDE.

 - We cite [Salimans2022] and acknowledge that the 3 learning objectives are equivalent to the ones we propose in our Table 1.

 - We discuss the image-to-image diffusion models in Section 7 such as [Zhao2022] and [Meng2021].

 - We numbered the theorem and propositions.

 - We replaced the density notation $f$ by $p$.

 - We fixed the minor typos and wording reported by the reviewers.

We answered the reviewer's specific questions and comments individually. We note that Reviewer W2Pp has lowered his score from 5 to 3 such that the average score went from 6.25 to 5.75. The exchange with Reviewer W2Pp spans multiple points regarding what is expected from a paper. Our main objective with this paper is to provide an alternative approach to deterministic denoising diffusion that readers will find simple and clear. Hence, the most important divergences with Reviewer W2Pp are

> However, I disagree with the authors that it's a "simpler entry point".

> As a practitioner in diffusion models, I feel that the forward-backward SDE framework in [1, 2] is clearer than the blend/deblend perspective.

>  In addition, I wouldn't call those linear/semi-linear forward ODE/SDEs ``advanced math".

We definitely respect that Reviewer W2Pp prefers the SDE formalism and finds it simple. We can hardly convince him otherwise since it is a matter of personal preference. Nonetheless, we believe that there is an audience that will benefit from our derivation and appreciate its simplicity and originality. With our paper, readers with undergrad calculus and probability knowledge can now derive a DDIM-like ODE. Furthermore, exploring alternative derivations can be fruitful. Our sampling interpretation of (de)blending is novel in the field and might lead to new connections and insights.

We leave it to the reviewer's discussion to decide whether the paper reaches the objective of providing a novel, simple and interesting take on the topic.

---

### Decision · Program_Chairs · 2023-01-20

**Decision:**

Reject

**Justification For Why Not Higher Score:**

The main weaknesses are: (1) The proposed idea is not entirely new and can be seen as a special case of a recent method of Peluchetti et al. (2) The paper lacks some important baselines that can also map between arbitrary distributions, e.g. mapping the source distribution to a hemisphere and then mapping the hemisphere to the target distribution. (3) Moreover the empirical results are quite limited and only work when the source distribution is Gaussian.

**Justification For Why Not Lower Score:**

N/A

**Metareview: Summary, Strengths And Weaknesses:**

This paper introduces a method for mapping between arbitrary densities that avoids using Langevin dynamics or score-matching. Instead it is based on Bayes rule. However the proposed idea is not entirely new and can be seen as a special case of a recent method of Peluchetti et al. The paper also lacks some important baselines that can also map between arbitrary distributions, e.g. mapping the source distribution to a hemisphere and then mapping the hemisphere to the target distribution. Moreover the empirical results are quite limited and only work when the source distribution is Gaussian.

**Summary Of Ac-Reviewer Meeting:**

The reviews covered different weaknesses in the paper (omission of earlier work, missing baselines and weak empirical results), and after discussing and taking it all into account, the decision to reject was unanimous.